# A Robust Dead Reckoning Algorithm Based on Wi-Fi FTM and Multiple Sensors

**Yue Yu [1]**, **Ruizhi Chen [1,2,*]**, **Liang Chen [1]**, **Guangyi Guo [1]**, **Feng Ye [1]** and **Zuoya Liu [1]**

[1]   State Key Laboratory of Information Engineering in Surveying, Mapping and Remote Sensing (LIESMARS),
     Wuhan University, Wuhan 430079, China; yue.yu@whu.edu.cn (Y.Y.); l.chen@whu.edu.cn (L.C.);
     guangyi.guo@whu.edu.cn (G.G.); yefeng92@whu.edu.cn (F.Y.); zyliu0017@whu.edu.cn (Z.L.)
[2]   Collaborative Innovation Center of Geospatial Technology, Wuhan University, Wuhan 430079, China
[*]   Correspondence: ruizhi.chen@whu.edu.cn; Tel.: +86-027-87731869-801

**Abstract:** More and more applications of location-based services lead to the development of indoor positioning technology. Wi-Fi-based indoor localization has been attractive due to its extensive distribution and low cost properties. IEEE 802.11-2016 now includes a Wi-Fi Fine Time Measurement (FTM) protocol which provides a more robust approach for Wi-Fi ranging between the mobile terminal and Wi-Fi access point (AP). To improve the positioning accuracy, in this paper, we propose a robust dead reckoning algorithm combining the results of Wi-Fi FTM and multiple sensors (DRWMs). A real-time Wi-Fi ranging model is built which can effectively reduce the Wi-Fi ranging errors, and then a multisensor multi-pattern-based dead reckoning is presented. In addition, the Unscented Kalman filter (UKF) is applied to fuse the results of Wi-Fi ranging model and multiple sensors. The experiment results show that the proposed DRWMs algorithm can achieve accurate localization performance in line-of-sight/non-line-of-sight (LOS)/(NLOS) mixed indoor environment. Compared with the traditional Wi-Fi positioning method and the traditional dead reckoning method, the proposed algorithm is more stable and has better real-time performance for indoor positioning.

**Keywords:** Indoor positioning; Dead reckoning; Wi-Fi Fine Time Measurement; Multiple sensors; Unscented Kalman filter

## 1. Introduction

In Global Navigation Satellite System (GNSS)-denied indoor environments, various indoor localization systems based on different techniques, such as ultra- wideband (UWB) [1], bluetooth [2], Wi-Fi [3], light source [4], and multi-sensors [5] have been developed for location-based services (LBS).

Wi-Fi-based indoor localization has been more attractive compared with other signals due to its extensive application and low cost properties. Multiple characteristics estimated from Wi-Fi signals can be used for indoor localization such as received signal strength indication (RSSI) [6], channel impulse response (CIR) [7], time of arrival (TOA) [8], angle of arrival (AOA) [9], and channel state information (CSI) [10]. Other Wi-Fi-based localization techniques such as multi-source fusion [11] and fingerprinting [12] can also be used in complex indoor scenarios. In 2016, IEEE 802.11 standardized the Fine Time Measurement (FTM) protocol which can provide meter-lever localization accuracy according to the Wi-Fi alliance [13]. Ibrahim M and his partners analyzed the key factors and parameters which affect the Wi-Fi ranging performance based on the open platform and revisited standard error correction techniques for a Wi-Fi FTM-based localization system [14].

However, when giving a complex indoor scenario where the direct transmission path between the transceivers is blocked, the distance errors measured by Wi-Fi FTM cannot be easily eliminated due to the lack of a line-of-sight (LOS) path [15]. To make it worse, the measurement errors are in

different statistics in different indoor scenarios, such as office, corridor, and underground parking [16]. In addition to signal-based indoor positioning technology, inertial localization technology such as pedestrian dead reckoning (PDR) is also widely used in indoor scenarios where the signal source is insufficient due to its passivity, autonomy, and short-term precision [17]. Recently, multi-source fusion positioning and combined navigation have attracted extensive attention because of their high precision and stability which can also be used in indoor localization [18–20]. Several fusion methods such as the Kalman filter (KF) [21], Extended Kalman filter (EKF) [22], and Particle filter (PF) [23] can effectively integrate different data sources to get more accurate positioning results.

This paper proposes a robust dead reckoning algorithm based on Wi-Fi FTM and multiple sensors (DRWMs). The contribution of this work is summarized as follows:

(1)　To improve the traditional multi-sensor-based dead reckoning method, a multi-pattern-based step detection and location updating algorithm is proposed in order to adapt to complex indoor walking modes.
(2)　A real-time ranging model based on Wi-Fi FTM is presented which can effectively reduce the Wi-Fi ranging error caused by clock deviation, non-line-of-sight (NLOS), and multipath propagation.
(3)　Based on the fusion of Wi-Fi ranging model and multi-pattern-based dead reckoning method, DRWMs is proposed. The combination of the real-time Wi-Fi FTM ranging model and the multi-sensor estimation method effectively improves the accuracy and stability of final dead reckoning.

The rest of this paper is organized as follows. Section 2 introduces the theoretical framework about the principle of Wi-Fi FTM-based indoor localization method and designs a multi-pattern-based dead reckoning algorithm using multiple sensors. Section 3 builds a real-time Wi-Fi ranging model which can effectively reduce the error of clock deviation, NLOS, and multipath propagation. Section 4 presents the DRWMs algorithm using Unscented Kalman filter (UKF) to fuse the results of the Wi-Fi ranging model and multiple sensors. Section 5 describes the experimental results of the proposed algorithm. Section 6 will conclude this paper and point out the future work.

## 2. Theoretical Framework

Traditional Wi-Fi-based positioning methods usually use RSSI to calculate the distance between the mobile terminal and Wi-Fi AP or use the fingerprint method to get the location of a pedestrian. Compared with the Wi-Fi FTM, the RSSI fingerprinting-based Wi-Fi positioning method is much more dependent on the environment because of the multipath propagation. RSSI is also highly interference dependent, hardware dependent, and sensitive to environmental factors such as temperature [24–26]. In addition, the RSSI-based Wi-Fi positioning method has relatively poor real-time performance; it may take one or more seconds to get a new location by scanning nearby AP in all bands. Wi-Fi FTM uses real time measurement based on clocks, by which we can get a much higher update rate of 3Hz or more and a fine measurement result with pico-seconds-based fine-grained time recognition. The traditional dead reckoning method provides the relative position of the pedestrian by calculating the real-time heading difference and step length. It can only be used in case of walking forward; errors are cumulated when other walking patterns occur, because the heading difference remains the same as the forward walking pattern. In this section, a more robust Wi-Fi-based positioning method and multi-pattern-based dead reckoning method are proposed.

### 2.1. Positioning Method Based on Wi-Fi FTM

Wi-Fi FTM protocol enables distance ranging between the mobile terminal and Wi-Fi access point (AP). The whole procedure is described as follows. First, the mobile terminal sends an FTM request to the Wi-Fi AP. Then, the Wi-Fi AP receives the request and returns an ACK signal to the mobile terminal. After that, several FTM feedbacks are sent from the Wi-Fi AP to the mobile terminal, and, then, the mean round-trip time (RTT) can be calculated. This process can also be performed between

several mobile terminals and Wi-Fi APs at the same time. Figure 1 shows the whole procedure in which the parameter names 'FTMs per Burst' can be changed to improve the FTM accuracy by multiple measurements. The mean RTT information within one period is calculated by Equation (1):

$$\text{RTT} = \frac{1}{n} \cdot \sum_{k=1}^{n} ([t_4(k) - t_1(k)] - [t_3(k) - t_2(k)]) \tag{1}$$

where $t_1(k)$ is the timestamp when the FTM framework first sent by Wi-Fi AP, $t_2(k)$ is the timestamp when the FTM signal arrives to the mobile terminal, $t_3(k)$ is the timestamp when the mobile terminal returns the ACK signal to Wi-Fi AP, $t_4(k)$ is the timestamp when the ACK signal is finally received by the Wi-Fi AP, and the parameter $n$ is the number of FTMs per burst among one ranging period. Generally, the protocol excludes the processing time on the mobile terminal by subtracting $(t_3(k) - t_2(k))$ from the total round-trip time $(t_4(k) - t_1(k))$, which represents the time from the instant the FTM message is being sent $(t_1(k))$ to the instant the ACK is being received $(t_4(k))$. This calculation is repeated for each FTM-ACK exchange, and the final RTT is the average over the number of FTMs per burst. In this paper, the parameter FTMs per burst $n$ is set as 30 to minimize the measurement noise, so it will take more time to complete this procedure compared to a smaller FTMs per burst [14]. The sampling rate of the RTT depends on the hardware performance of the processor and bandwidth of Wi-Fi, and the frequency of the processor should satisfy pico-seconds fine-grained requirements according to [13].

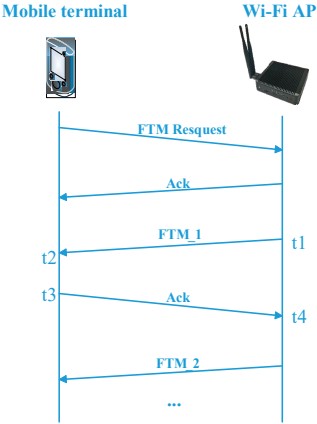

**Figure 1.** Procedure of Wi-Fi Fine Time Measurement (FTM)

The distance between mobile terminal and Wi-Fi AP can be calculated by Equation (2):

$$D_{\text{RTT}} = \frac{C}{2n} \cdot \sum_{k=1}^{n} ([t_4(k) - t_1(k)] - [t_3(k) - t_2(k)]) \tag{2}$$

where $C$ is the speed of light and $D_{\text{RTT}}$ is the final distance calculated by one FTM period.

The least squares (LS) algorithm [27] can be used for real-time localization after receiving distance information from three or more APs:

$$x_p = (A^T A)^{-1} A^T b$$

$$x_p = \begin{bmatrix} x & y \end{bmatrix}^T, A = 2 \cdot \begin{bmatrix} x_1 - x_2 & y_1 - y_2 \\ x_1 - x_3 & y_1 - y_3 \\ \vdots & \vdots \\ x_1 - x_j & y_1 - y_j \end{bmatrix}, b = \begin{bmatrix} D_{\text{RTT}}(2) - D_{\text{RTT}}(1) - (x_2^2 - x_1^2) - (y_2^2 - y_1^2) \\ D_{\text{RTT}}(3) - D_{\text{RTT}}(1) - (x_3^2 - x_1^2) - (y_3^2 - y_1^2) \\ \vdots \\ D_{\text{RTT}}(j) - D_{\text{RTT}}(1) - (x_j^2 - x_1^2) - (y_j^2 - y_1^2) \end{bmatrix} \tag{3}$$

where $x_p$ is the localization result, $j$ is the number of APs, $x_j$ and $y_j$ are the position of Wi-Fi AP, and $D_{\text{RTT}}(j)$ is real-time RTT data received from each Wi-Fi AP. It should be noted that the number of APs

needed to solve the above equation is at least one more than the dimension of the $x_p$. In theory, we can get more accurate positioning results by increasing the number of APs [28].

### 2.2. Multi-Pattern-Based Dead Reckoning via Multiple Sensors

Mobile phones are generally equipped with multiple MEMS (Micro-Electro-Mechanical System) sensors such as an accelerometer, a magnetometer, and a gyroscope. When pedestrians carrying mobile terminals, these sensors can reflect the movement. Therefore, the sensor data can be used for indoor localization. The walking patterns of pedestrians can significantly affect the accuracy of positioning results which should be detected. The multi-pattern-based PDR algorithm proposed in this paper mainly contains the following two steps: (1) Multi-pattern-based step detection and step-length estimation; (2) location updates.

#### 2.2.1. Multi-Pattern-Based Step Detection and Step-Length Estimation

The basic principle of step number measurement is that the vertical value of acceleration data is periodically changing when walking. The number of steps can be used to update the position combining step-length and heading difference. The original data from the accelerometer contains noise which should be smoothed. A moving average filter can be used to reduce the noise, and the obvious step character can be extracted after filtering [21].

Different patterns may exist between walking procedures such as walking forward, walking backward, and lateral walking, which may cause coordinate updating error without detection. In case of walking forward, steps can be detected by a peak and valley detection algorithm [29]. However, this step detection algorithm should be modified in the case of other walking patterns. In this part, a multi-pattern-based step detection algorithm is proposed to realize real-time step detection and step-length calculation based on different walking patterns.

In the step detection process with higher real-time requirements, time-domain features of acceleration can be used to identify the walking pattern. The handheld posture of mobile phone is shown in Figure 2, which is cited in [30].

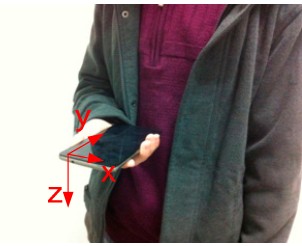

**Figure 2.** Handheld posture of mobile phone.

Acceleration shows periodic waveform characteristics when a pedestrian walks. In the case of walking forward and walking backward, the Y-axis is seen as the walking axis, and the slope of data from the Y-axis is extracted to identify the walking pattern by combining peak and valley detection of the Z-axis. The slope of the acceleration data from Y-axis defined in Equation (4).

$$\left(\begin{array}{l} K_1 = A_y(m_1) - A_y(m_1 - 1) \\ K_2 = A_y(m_1 + 1) - A_y(m_1) \end{array}\right. \tag{4}$$

where $m_1$ is the timestamp when the peak or valley of Z-axis value is detected during one step period and $A_y$ is the acceleration data from Y-axis.

Figure 3 shows the comparison of walking forward and backward; the tester walked forward for 8 m and then walked backward to original point. The sample rate of accelerometer is 50 Hz, and the raw data is filtered by moving average filter with a width of 10. Figure 4 is the same. The red dots

show the peaks or valleys of the Z-axis data as corresponding to the red stars, which represent the Y-axis data at the moment $m_1$. When the conditions $K_1 > 0$ and $K_2 > 0$ are met, in the case of walking forward, $m_1$ is the timestamp when the peak of Z-axis value is detected. In case of walking backward, $m_1$ is the timestamp when the valley of the Z-axis value is detected.

In case of left lateral walking and right lateral walking, the X-axis is seen as the walking axis. The data slope of the X-axis is extracted to identify the walking pattern by combining the peak and valley detections of the Z-axis. The slope of acceleration data from X-axis defined in Equation (5).

$$\left( \begin{array}{l} K_3 = A_x(m_2) - A_x(m_2 - 1) \\ K_4 = A_x(m_2 + 1) - A_x(m_2) \end{array} \right. \tag{5}$$

where $m_2$ is the timestamp when the peak of Z-axis value is detected during one step period and $A_x$ is the acceleration data from X-axis. Figure 4 shows the comparison of left lateral walking and right lateral walking. The tester walked left laterally for 6 m and then walked right laterally to the original point. The red dots show the peaks or valleys of the Z-axis data, as corresponding to the red stars, which represent the X-axis data at the moment $m_2$. When the conditions $K_3 < 0$ and $K_4 < 0$ are met, in case of left lateral walking, $m_2$ is the timestamp when the peak of the Z-axis value is detected. When the conditions $K_3 > 0$ and $K_4 > 0$ are conformed, in case of right lateral walking, $m_2$ is the timestamp when the peak of the Z-axis value is detected. Only a texting pattern is considered in this work; as shown in Figure 2, subtle tilt does not affect the recognition results because the Equations (4) and (5) calculate the changing trend of the acceleration data.

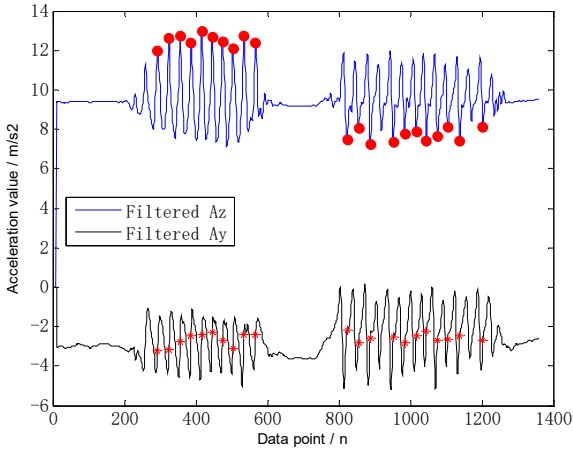

**Figure 3.** Comparison of walking forward and backward.

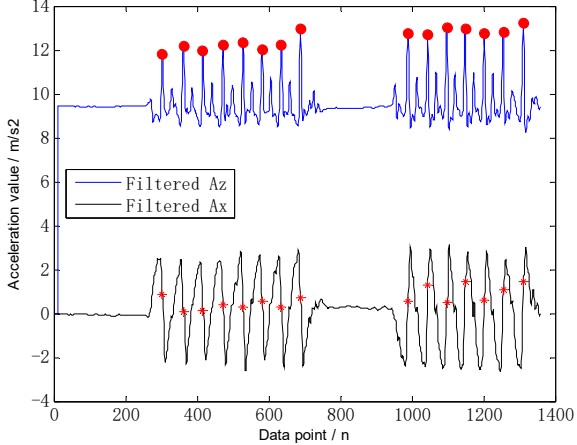

**Figure 4.** Comparison of lateral walking.

After the multi-pattern-based step detection is completed, the Weinberg model is used to calculate the real-time step length [31], described as follows:

$$\text{step\_length} = K \sqrt[4]{A_{\max} - A_{\min}} \tag{6}$$

where $A_{\max}$ and $A_{\min}$ are the maximum and minimum values of the Z-axis acceleration during one step period and $K$ is the ratio of the real and the estimated distance:

$$K = \frac{d_{\text{real}}}{d_{\text{estimated}}} \tag{7}$$

### 2.2.2. Location Update

A traditional dead reckoning algorithm is used to calculate the real-time position coordinates of a pedestrian in the case of walking forward. In the cases of walking backward, left lateral walking, and right lateral walking—since the pedestrian's heading angle remains unchanged—the positioning trajectory will continue updating forward while the true trajectory should be updated in the opposite or perpendicular direction to avoid causing the accumulation of positioning errors. By using the multi-pattern step detection algorithm, different walking patterns are detected, and the corresponding step length is calculated. Using a location updating algorithm based on multiple walking patterns can effectively reduce the cumulative error of dead reckoning.

For the backward walking pattern, as shown in Figure 5, the pedestrian starts from the original location, continuously walks to point A and point B, and then walks backwards from point B to point A. The final position coordinates of the pedestrian measured by the traditional dead reckoning (DR) algorithm is the D point, which is opposite to the actual walking trajectory of the pedestrian. In case of either a left lateral walking pattern or right lateral walking pattern, the pedestrian walks from point A to point B and then walks laterally to point $C_1$ and $C_2$. When using the traditional DR algorithm, the walking trajectory still shows from point B to point D, perpendicular to the actual trajectory of the pedestrian.

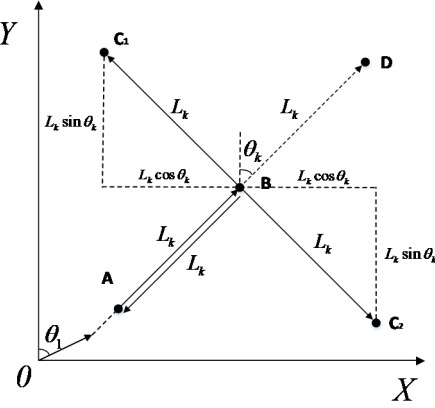

**Figure 5.** Multi-pattern dead reckoning.

For the above four walking patterns, the multi-pattern-based location updating equation is described as follows:

$$\left( \begin{array}{l} X(k) = X(k-1) + S_k L_k \sin\theta_k + U_k L_k \cos\theta_k \\ Y(k) = Y(k-1) + S_k L_k \cos\theta_k - U_k L_k \sin\theta_k \end{array} \right. \tag{8}$$

where $L_k$ is the step-length and $\theta_k$ is the difference of heading angle compared with initial heading which is provided by the calibrated magnetometer [32]. The real-time heading is calculated by an EKF filter which combines the outputs of the gyroscope and magnetometer [30]. A threshold $\Delta h$ is used to eliminate the heading jitter when walking and keep the heading unchanged when the

heading increment is smaller than $\Delta h$. $S_k$ is the flag of detected walking patterns, including walking forwards and backwards. $S_k$ is set to 1 when a walking forwards pattern is detected, while $S_k$ is set to −1 when a walking backwards pattern detected. In case of other walking patterns, $S_k$ is set to 0. $U_k$ is the flag of lateral walking patterns. $U_k$ is set to −1 when a left lateral walking pattern detected, and $U_k$ is set to 1 when a right lateral walking pattern detected. In case of other walking patterns, $U_k$ is set to 0. The real-time location of pedestrians can be accurately updated using the above equation. Compared with the traditional location updating method provided in [30], when backwards or lateral walking patterns happen, the proposed location updating algorithm can effectively classify the different walking patterns and go on to decrease the positioning error caused by misjudgment.

### 2.3. Challenges of Indoor Positioning for Pedestrians

The Wi-Fi FTM protocol included in IEEE 802.11-2016 enables Wi-Fi ranging between the mobile terminal and Wi-Fi AP and provides a new direction for Wi-Fi-based indoor positioning. PDR is based on multiple sensors and calculates the step length and heading difference to update the relative coordinates of pedestrian. These two methods are the focus of recent research. However, in summarizing the work of predecessors, this paper found that the Wi-Fi FTM-based approach and PDR algorithm continue to face the following challenges:

1.　Clock Deviation of Wi-Fi FTM

IEEE 802.11-2016 gives a definition about Wi-Fi ranging which supports exchange of FTM information and timestamps captured between the mobile terminal and Wi-Fi AP [13]. TOA and DOA methods [33,34] are included in parameters of each single FTM exchange. The clock-based timestamps capture cannot be absolutely precise; thus, one of the most important factors that causes the Wi-Fi ranging error is the clock deviation error, which is caused by an initial deviation and a random error, which are inconsistent with different mobile terminals and Wi-Fi APs. Generally, the initial deviation can be calibrated by calculating the difference between the true value and the measurement in field tests [14]. The random clock error causes a fluctuation of the received RTT data on the mobile terminal, which can be eliminated by filters such as the Kalman Filter (KF) [35], Mean filter [36], Gaussian filter [37].

2.　NLOS and Multipath Propagation

In a complex indoor environment, the Wi-Fi ranging results may contain NLOS distance which may not be easily detected. The existence of NLOS errors will significantly degrade the localization performance, and, hence, the mitigation of NLOS errors becomes an urgent task [38,39]. In indoor scenarios, the direct path between transceivers are easily blocked due to the complexity of the indoor layout. The propagation of the waves' path may lengthen due to reflection, refraction, and scattering, all of which can cause large positioning errors for indoor navigation with ranged-based methods [40,41].

3.　Cumulative Error of Inertial Sensors

In the PDR method, the current location is based on the continuous reckoning of the previous locations and pedestrian steps. The step-length error and step-count deviation caused by cumulative error of inertial sensors will influence positioning accuracy [42]. In addition, heading drift is also an important factor [43]. In [44], it is noted that the cumulative error is always existing and growing with time, while the requirement of indoor localization is within 5 m. Therefore, it is necessary to eliminate the cumulative error.

### 3. Ranging Model of Wi-Fi FTM

IEEE 802.11-2016 enhanced Wi-Fi ranging by providing Wi-Fi FTM—any mobile terminals which support this new protocol can use this function. However, the raw data from FTM contains a clock

deviation error and could be influenced by NLOS and multipath propagation. In this section, a real-time ranging model which can effectively reduce the error caused by clock deviation, NLOS, and multipath is proposed.

### 3.1. Model of Clock Deviation Error

It can be noticed from Figure 1 that the final RTT data is calculated by the different timestamps carried by the Wi-Fi FTM framework. In other words, the detection precision of the timestamps determines the accuracy of the calculated RTT. However, these timestamps are not the true time instant when the signals arrive or leave the Wi-Fi AP and mobile terminal due to the procedure of signal processing and hardware delay. The true RTT data finally calculated by the Wi-Fi AP contains an initial time difference called $\Delta t_{\text{delay}}$, which is inconsistent between different hardware structures and processing methods of the signal—similar as TOA and DOA technology. $\Delta t_{\text{delay}}$ in one period of RTT can be described as follows:

$$
\begin{aligned}
\text{RTT}_{\text{true}} &= \left( \left[ \left( t_4 - t_{4\_\text{delay}} \right) - \left( t_1 + t_{1\_\text{delay}} \right) \right] - \left[ \left( t_3 + t_{3\_\text{delay}} \right) - \left( t_2 - t_{2\_\text{delay}} \right) \right] \right) \\
&= \text{RTT}_{\text{measurement}} - \left( t_{4\_\text{delay}} + t_{1\_\text{delay}} \right) - \left( t_{3\_\text{delay}} + t_{2\_\text{delay}} \right) \\
&= \text{RTT}_{\text{measurement}} - \Delta t_{\text{delay}}
\end{aligned}
\tag{9}
$$

where $\text{RTT}_{\text{true}}$ is the true ranging result after subtracting $\Delta t_{\text{delay}}$ from real-time measurement result $\text{RTT}_{\text{measurement}}$—t is defined in Equation (1).

The random error of the clock exists during the each FTM processing due to the status of the system and signal propagation in different environments. When the distance is finally calculated, it fluctuates within a range due to random error. In general, $\Delta t_{\text{random}}$ can be assumed as Gaussian-distributed variables with zero mean and variance after calibration, which is described in Equation (10):

$$
\Delta t_{\text{random}}(m) = \frac{1}{\sqrt{2\pi}\sigma} \cdot \exp(-\frac{m^2}{2\sigma^2})
\tag{10}
$$

where $m$ is the timestamp.

With the understanding of the initial clock deviation and random error of the clock, RTT in one period can be described as follows:

$$
\text{RTT} = (t_4 - t_1) - \left( t_3 - t_2 \right) + \Delta t_{\text{delay}} \Delta t_{\text{random}}
\tag{11}
$$

where $\Delta t_{\text{delay}}$ exists before ranging which can't be easily detected directly by hardware. As such, before using the Wi-Fi ranging system, some calibration measurements have to be done to eliminate the initial clock deviation $\Delta t_{\text{delay}}$. $\Delta t_{\text{random}}$ exists during the ranging process which can result in signal fluctuation. Both errors cannot be fully eliminated by calibration and filter, so a further processing algorithm is needed.

### 3.2. Model of NLOS and Multipath Propagation

It is important to consider an indoor environment with several available APs which support the Wi-Fi FTM. In the case that the mobile terminal is moving, parts of APs may be blocked in a short time that cause the NLOS and multipath propagation which may not be detected by themselves. In order to solve the problem, a ranging model containing the effects of NLOS and multipath propagation is proposed. Locations of Responders/APs is indicated as $P_i$, and location of the mobile terminal is indicated as $P$. Taking the effect of NLOS and multipath into consideration can result in the following model:

$$
\begin{aligned}
L_i &= L_0 + \|P - P_i\| + e_i + d_{\text{random}} \\
d_{\text{random}} &= \frac{C \cdot \Delta t_{\text{random}}}{2}
\end{aligned}
\tag{12}
$$

where $L_i$ is the measured distance, $i$ is used to differentiate different APs, $L_0$ is the extra ranging distance caused by multipath, $\boldsymbol{P} = [x_0 \ y_0]^T$ indicates the location of mobile terminal, $\boldsymbol{P}_i$ is the location of Wi-Fi AP, $\|\boldsymbol{P} - \boldsymbol{P}_i\|$ is the matrix norm that indicates the Euclidean distance between mobile terminal and Wi-Fi AP, $e_i$ is the NLOS error which indicates the difference between the final propagation distance of the signal and the true straight line distance when LOS path is lacked [45], $d_{\mathrm{random}}$ is the random error of measurements which confront to a zero-mean Gaussian distribution with a variance of 0.25 [14], and $\Delta t_{\mathrm{random}}$ is defined in Equation (10). We assume that $e_i$ is much larger than $d_{\mathrm{random}}$, with a boundary of $b_i$, which always indicates the distance between mobile terminal to the farthest AP. Since $d_{\mathrm{random}}$ is always between $-0.5$m and $0.5$m according to [14], we move $e_i$ to the left side, square both sides, and neglect the $d^2_{\mathrm{random}}$:

$$(L_i - e_i)^2 \approx (L_0 + \|\boldsymbol{P} - \boldsymbol{P}_i\|)^2 + 2d_{\mathrm{random}}(L_0 + \|\boldsymbol{P} - \boldsymbol{P}_i\|) \tag{13}$$

Then $d_{\mathrm{random}}$ can be obtained as:

$$d_{\mathrm{random}} \approx \frac{(L_i - e_i)^2 - (L_0 + \|\boldsymbol{P} - \boldsymbol{P}_i\|)^2}{2(L_0 + \|\boldsymbol{P} - \boldsymbol{P}_i\|)} \tag{14}$$

Then define a function on $e_i$:

$$f(e_i) = \frac{|(L_i - e_i)^2 - (L_0 + \|\boldsymbol{P} - \boldsymbol{P}_i\|)^2|}{L_0 + \|\boldsymbol{P} - \boldsymbol{P}_i\|} \tag{15}$$

The LS [45] algorithm can be used to solve Equation (15) with the condition $0 < e_i < b_i$. Assuming that the number of available Wi-Fi APs is $N$:

$$\min_{\boldsymbol{P}, L_0} \max_{e_i} \sum_{i=1}^{N} \frac{f^2(e_i)}{4\sigma^2} = \min_{\boldsymbol{P}, L_0} \sum_{i=1}^{N} \frac{[\max_{e_i} f(e_i)]^2}{4\sigma^2} \tag{16}$$

With the condition $0 < e_i < b_i$, $\max_{e_i} f(e_i)$ can be divided into two cases:
First case: $L_i <= b_i$, then $\max_{e_i} f(e_i) = \max\{f(0), f(L_i), f(b_i)\}$.
Second case: $L_i > b_i$, then $\max_{e_i} f(e_i) = \max\{f(0), f(b_i)\}$.
Then Equation (16) can be translated as:

$$\min_{\boldsymbol{P}, L_0, \{\rho_i\}} \sum_{i=1}^{N} \rho_i$$
$$s.t. \frac{f^2(0)}{4\sigma^2} \le \rho_i, \ \frac{f^2(L_i)}{4\sigma^2} \le \rho_i, \ \frac{f^2(b_i)}{4\sigma^2} \le \rho_i (L_i \le b_i) \tag{17}$$

Introducing variables $y = \|\boldsymbol{P}\|^2$, $r = L_0{}^2$, $k_i = 2L_0\|\boldsymbol{P} - \boldsymbol{P}_i\|$ creates the following equation:

$$\min_{\boldsymbol{P}, L_0, y, r, \{\rho_i, k_i\}} \sum_{i=1}^{N} \rho_i$$
$$s.t. \frac{\left(L_i{}^2 - y - r + 2\boldsymbol{P}_i{}^T\boldsymbol{P} - \|\boldsymbol{P}_i\|^2 - k_i\right)^2}{y + r - 2\boldsymbol{P}_i{}^T\boldsymbol{P} + \|\boldsymbol{P}_i\|^2 + k_i} \le 4\sigma^2\rho_i,$$
$$\frac{\left(k_i{}^2 - 2L_i k_i + L_i{}^2 - y - r + 2\boldsymbol{P}_i{}^T\boldsymbol{P} - \|\boldsymbol{P}_i\|^2 - k_i\right)^2}{y + r - 2\boldsymbol{P}_i{}^T\boldsymbol{P} + \|\boldsymbol{P}_i\|^2 + k_i} \le 4\sigma^2\rho_i,$$
$$\frac{\left(2\boldsymbol{P}_i{}^T\boldsymbol{P} - \|\boldsymbol{P}_i\|^2 - k_i - y - r\right)^2}{y + r - 2\boldsymbol{P}_i{}^T\boldsymbol{P} + \|\boldsymbol{P}_i\|^2 + k_i} \le 4\sigma^2\rho_i,$$
$$y = \|\boldsymbol{P}\|^2, r = L_0{}^2, k_i = 2L_0\|\boldsymbol{P} - \boldsymbol{P}_i\|(L_i \le b_i) \tag{18}$$

Based on the assumption that $e_i >> |d_{random}|$, Equation (18) can be transformed into a more tightened problem:

$$A[P^T, y, L_0, r]^T \leq f$$
$$A = \begin{bmatrix} -2P_1^T & 1 & 2L_1 & -1 \\ \vdots & \vdots & \vdots & \vdots \\ -2P_N^T & 1 & 2L_N & -1 \end{bmatrix}, f = \begin{bmatrix} L_1{}^2 - \|P_1\|^2 \\ \vdots \\ L_N{}^2 - \|P_N\|^2 \end{bmatrix} \tag{19}$$

Problem (19) is non-convex. With the constraints $y = \|P\|^2, r = L_0{}^2$, the commonly used standard second-order cone relaxation technique can be applied to relax them as $\|P\|^2 \leq y$ and $L_0{}^2 \leq r$. For the constraint $k_i = 2L_0\|P - P_i\|(L_i \leq b_i)$, transforming the equation to a convex one can result in the following constraint:

$$0 \leq k_i = 2L_0\|P - P_i\| \leq r + y - 2P_i^T P + \|P_i\|^2$$
$$k_i{}^2 = 4L_0{}^2\|P - P_i\|^2 \leq 4r(y - 2P_i^T P + \|P_i\|^2) \tag{20}$$

Utilizing the relaxations for constraints and the approximations in Equation (19), a convex second-order cone program is as follows:

$$\min_{P, L_0, y, r, \{\rho_i, k_i\}_{i=1}^N} \sum_{i=1}^N \rho_i$$
$$s.t. \frac{\left(L_i{}^2 - y - r + 2P_i^T P - \|P_i\|^2 - k_i\right)^2}{y + r - 2P_i^T P + \|P_i\|^2 + k_i} \leq 4\sigma^2 \rho_i,$$
$$\frac{\left(k_i{}^2 - 2L_i k_i + L_i{}^2 - y - r + 2P_i^T P - \|P_i\|^2 - k_i\right)^2}{y + r - 2P_i^T P + \|P_i\|^2 + k_i} \leq 4\sigma^2 \rho_i, \tag{21}$$
$$\frac{\left(2P_i^T P - \|P_i\|^2 - k_i - y - r\right)^2}{y + r - 2P_i^T P + \|P_i\|^2 + k_i} \leq 4\sigma^2 \rho_i,$$
$$\|P\|^2 \leq y, L_0{}^2 \leq r, (19), (20)$$

The optimal estimated value of $\|P - P_i\|$ can be gotten from the above formulas and constraints, which approximate to the true value of RTT data from each Wi-Fi AP.

## 4. Integrated Localization Based on Wi-Fi FTM and PDR

This work provides multiple sensors and a Wi-Fi ranging model to estimate pedestrian position. The two methods have their own advantages and disadvantages. The positioning method based on Wi-Fi FTM can provide the exact coordinates directly, but it is affected by NLOS and multipath propagation. Multi-sensor estimation is more accurate within a short distance, but there exist cumulative errors which cannot be used for a long time. This paper combines the two methods to overcome the shortage of each. The estimation based on Wi-Fi FTM could eliminate the cumulative error caused by multi-sensor estimation and could provide an initial location to UKF model. Multi-sensor estimation has good time-recursive performance and can decrease error of NLOS and multipath propagation caused by Wi-Fi FTM. Since UKF does not ignore high-order terms, it has higher filtering accuracy than EKF under the same conditions, and the computational complexity is smaller than PF; thus, the UKF is selected for data fusion in this paper which combines the proposed two methods to achieve higher positioning accuracy. The whole framework of proposed DRWMs algorithm is shown in Figure 6.

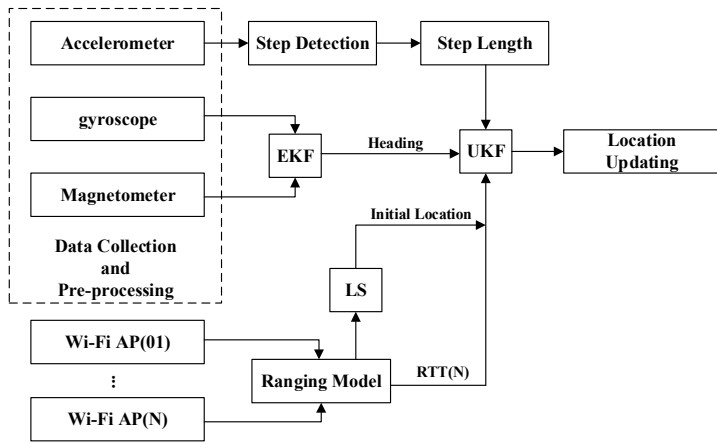

**Figure 6.** Framework of the Wi-Fi FTM and multiple sensors (DRWM)s algorithm.

### 4.1. System Model Based on Unscented Kalman filter

In the process of fusion, the difference of position estimated by multiple sensors is calculated as predicted value, and the distance calculated by the Wi-Fi FTM-based ranging model is used as the observed value. The system state equation is as follows:

$$X(t) = \begin{bmatrix} x(t) \\ y(t) \end{bmatrix} = \psi \begin{bmatrix} x(t-1) \\ y(t-1) \end{bmatrix} + BS(t) + v \tag{22}$$

where $[x(t-1)\ y(t-1)]^T$ are the 2D coordinates of the pedestrian at time $t-1$, $S(t)$ is the real-time step-length, $v$ is the Gaussian noise with a noise matrix $Q$, $\psi$ is a unit matrix, and $B = [S_t \sin\theta_t + U_t \cos\theta_t\ S_t \cos\theta_t - U_t \sin\theta_t]^T$, indicates the difference of heading angle compared with initial heading. $S_k$ and $U_k$ are defined in Equation (6) and are used for multi-pattern step detection. The initial location $(x(0), y(0))$ is provided by the LS algorithm, which is defined in Equation (3).

The distance between the mobile terminal and each Wi-Fi AP is calculated by Wi-Fi FTM and is called the round-trip time (RTT); it can be used as an observed value after pre-processing by the ranging model proposed in Section 3. The accuracy of position estimation can be improved using multiple Wi-Fi APs. The observation equation is as follows:

$$Z(t) = \begin{bmatrix} \sqrt{(x(t) - x_1)^2 + (y(t) - y_1)^2} \\ \sqrt{(x(t) - x_2)^2 + (y(t) - y_2)^2} \\ \vdots \\ \sqrt{(x(t) - x_j)^2 + (y(t) - y_j)^2} \end{bmatrix} + U \tag{23}$$

where $U$ is the random error of Wi-Fi FTM with a noise matrix $R$, $U = [d^1_{\text{random}}, d^2_{\text{random}}, \cdots, d^j_{\text{random}}]^T$, and $d_{\text{random}}$ is defined in Equation (12). $x(t)$ and $y(t)$ are estimated by the system state equation. The Euclidean distance between the predicted position and each Wi-Fi AP is calculated as observation value, $j$ is the number of Wi-Fi AP, and $x_j$ and $y_j$ indicate the position of each AP.

### 4.2. Data Fusion via Unscented Kalman filter

The UKF is a well-known nonlinear state estimation method which shows superior performance at nonlinear estimation and data fusion [46,47]. The Unscented Kalman filter proposed in this paper is divided into eight steps:

(1)　Getting sigma point set based on the previous location $\overset{\wedge}{X}(t|t)$ and the corresponding weight:

$$
\begin{cases}
\overset{\eta=0}{\phantom{X}}\qquad\qquad\overset{\eta=1\sim\beta}{\phantom{X}}\qquad\qquad\overset{\eta=\beta+1\sim2\beta}{\phantom{X}} \\
X^{(\eta)}(t|t) = [\overset{\wedge}{X}(t|t), \overset{\wedge}{X}(t|t) + \sqrt{(\beta+\lambda)\boldsymbol{\phi}(t|t)}, \overset{\wedge}{X}(t|t) - \sqrt{(\beta+\lambda)\boldsymbol{\phi}(t|t)}] \\
w^{(0)} = \frac{\lambda}{\beta+\lambda}, \eta = 0 \\
w^{(\eta)} = \frac{\lambda}{2(\beta+\lambda)}, \eta = 1 \sim 2\beta
\end{cases}
\tag{24}
$$

where $\beta$ is the dimension of the state value, $\eta$ is the corresponding number of sigma point set, and $\lambda$ is the proportional parameter which is used to scale of the weight. $\boldsymbol{\phi}(t|t)$ is the state covariance matrix at the current moment $t$.

(2)　Further prediction of $2\beta + 1$ sigma point sets, $\eta = 0, 1, 2, \cdots, \beta + 1$:

$$
X^{(\eta)}(t+1|t) = \boldsymbol{\psi} X^{(\eta)}(t|t) + BS(t+1) + v
\tag{25}
$$

(3)　Weighting sigma point set, getting predicted value and covariance matrix.

$$
\overset{\wedge}{X}(t+1|t) = \sum_{\eta=0}^{2\beta} w^{(\eta)} X^{(\eta)}(t+1|t)
\tag{26}
$$

$$
\boldsymbol{\phi}(t+1|t) = \sum_{\eta=0}^{2\beta} w(\eta)[\overset{\wedge}{X}(t+1|t) - X^{(\eta)}(t+1|t)][\overset{\wedge}{X}(t+1|t) - X^{(\eta)}(t+1|t)]^T + Q
\tag{27}
$$

(4)　Getting the sigma point set again using UT transform based on the predicted state value.

$$
X^{(\eta)}(t+1|t) = [\overset{\wedge}{X}(t+1|t), \overset{\wedge}{X}(t+1|t) + \sqrt{(\beta+\lambda)\boldsymbol{\phi}(t+1|t)}, \overset{\wedge}{X}(t+1|t) - \sqrt{(\beta+\lambda)\boldsymbol{\phi}(t+1|t)}]
\tag{28}
$$

(5)　Further prediction of observation based on 2n + 1 sigma point sets of prediction, $\eta = 0, 1, 2, \cdots, 2\beta + 1$.

$$
Z^{(\eta)}(t+1|t) =
\begin{bmatrix}
\sqrt{(x^{(\eta)}(t+1|t) - x_0)^2 + (y^{(\eta)}(t+1|t) - y_0)^2} \\
\sqrt{(x^{(\eta)}(t+1|t) - x_2)^2 + (y^{(\eta)}(t+1|t) - y_2)^2} \\
\vdots \\
\sqrt{(x^{(\eta)}(t+1|t) - x_j)^2 + (y^{(\eta)}(t+1|t) - y_j)^2}
\end{bmatrix}
\tag{29}
$$

where $x^{(\eta)}(t+1|t)$ and $y^{(\eta)}(t+1|t)$ are calculated in $X^{(\eta)}(t+1|t)$.

(6)　Weighting sigma point sets, getting predicted observation value, and corresponding covariance matrix.

$$
\overset{\wedge}{Z}(t+1|t) = \sum_{\eta=0}^{2\beta} w^{(\eta)} Z^{(\eta)}(t+1|t)
\tag{30}
$$

$$
\boldsymbol{\phi}_{z_t z_t} = \sum_{\eta=0}^{2\beta} w(\eta)[Z^{(\eta)}(t+1|t) - \overset{\wedge}{Z}(t+1|t)][Z^{(\eta)}(t+1|t) - \overset{\wedge}{Z}(t+1|t)]^T + R
\tag{31}
$$

$$
\boldsymbol{\phi}_{x_t z_t} = \sum_{\eta=0}^{2\beta} w(\eta)[X^{(\eta)}(t+1|t) - \overset{\wedge}{Z}(t+1|t)][X^{(\eta)}(t+1|t) - \overset{\wedge}{Z}(t+1|t)]^T + R
\tag{32}
$$

where $\boldsymbol{\phi}_{z_t z_t}$ is the covariance matrix calculated by $\overset{\wedge}{Z}(t+1|t)$ and $Z^{(\eta)}(t+1|t)$, and $\boldsymbol{\phi}_{x_t z_t}$ is the covariance matrix calculated by $\overset{\wedge}{Z}(t+1|t)$ and $X^{(\eta)}(t+1|t)$.

(7) Calculating the Kalman gain.

$$K(t+1) = \boldsymbol{\phi}_{x_t z_t} \boldsymbol{\phi}_{z_t z_t}^{-1} \tag{33}$$

(8) System status and covariance updating.

$$\overset{\wedge}{\boldsymbol{X}}(t+1|t+1) = \overset{\wedge}{\boldsymbol{X}}(t+1|t) + \boldsymbol{K}(t+1)[\boldsymbol{Z}(t+1) - \overset{\wedge}{\boldsymbol{Z}}(t+1|t)] \tag{34}$$

$$\boldsymbol{\phi}(t+1|t+1) = \boldsymbol{\phi}(t+1|t) - \boldsymbol{K}(t+1)\boldsymbol{\phi}_{z_t z_t}\boldsymbol{K}^T(t+1) \tag{35}$$

The state equation describes the recursive relationship between input and output. The location updating equation for multiple sensors is used as the state equation. Compared with the 3 Hz sampling rate of Wi-Fi FTM, multi-sensor-based dead reckoning has better timing recursion because of much higher sampling rate of 50 Hz, which could be used as the state equation. The distance estimated by the Wi-Fi FTM-based ranging model is obtained as the observation. There is no cumulative error in the process of Wi-Fi FTM, so the estimation distance between the mobile terminal and each Wi-Fi AP can be used as the observed value. What one has to be aware of is that the UKF can only solve the random error, which could be seen as the Gaussian noise. In this paper, the Wi-Fi ranging model can effectively reduce the initial clock error, NLOS, and multipath effect, while random error—which is defined in Equation (10)—is added in the UKF model. Based on the covariance in the Unscented Kalman filter, this paper can obtain the corresponding weight of the Wi-Fi ranging model and the multi-sensor estimation and go on to get the most optimal position estimation.

To summarize, the proposed UKF fuses the estimation results of the two methods. On the one hand, the absolute position provided through the Wi-Fi FTM-based ranging model eliminates the cumulative error caused by multiple sensors; on the other hand, the multi-sensor recursive results for continuous trajectory reduce the effect of NLOS and multipath propagation caused by Wi-Fi FTM. Hence, a more accurate localization result is provided through the Unscented Kalman filter.

## 5. Experimental Results of DRWMs

In this section, several experiments are designed to verify the multi-pattern-based PDR algorithm, Wi-Fi FTM-based ranging model, and the proposed DRWMs algorithm. Two typical indoor environments were selected as the experimental sites. One contained a rectangular office and a long corridor; the other was part of a shopping mall. The Wi-Fi AP uses an Intel 8260 Wireless card and an Ubuntu 16.04 LTS as hardware and software platforms, which were custom-made by this work. A Google Pixel 3 was used as mobile terminal which supports Android P-based Wi-Fi FTM and can get real-time RTT data from surrounding Wi-Fi Aps. In addition, the Google Pixel 3 contained the sensors such as the accelerometer, gyroscope, and magnetometer. The sampling rates of the multi-sensor and Wi-Fi FTM was 50 Hz and 3 Hz, based on Google Pixel 3. The real-time location information calculated by UKF was acquired with frequency of 3 Hz. Timestamps within the multi-sensor method and Wi-Fi FTM were synchronized based on the time when RTT data returned. $\Delta h$ was set to 2.5° to eliminate the little heading jitter when walking. In the first experimental site, four APs were fixed on the stands with a 1.5m height in different locations of a rectangular office which contains NLOS and multipath propagation effects such as glass, partition shades, and a wall column. The tester could almost walk through the location of each AP, which ensured that the RTT data could be received anywhere in the office (12 m * 12 m). The position of each Wi-Fi AP is shown in Figure 7.

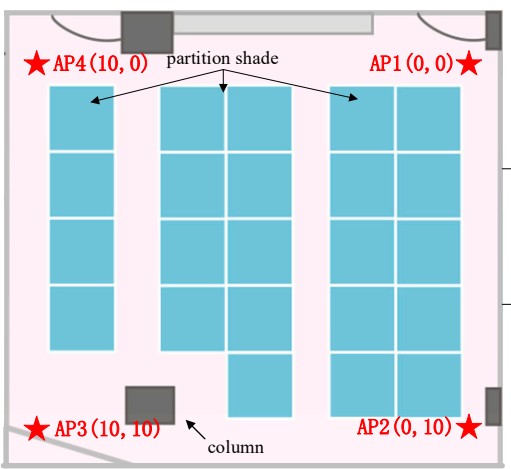

**Figure 7.** Deployment Wi-Fi access points (APs).

*5.1. Evaluation of Multi-Pattern-Based Dead Reckoning*

The estimation of steps and step-length acquired by multiple sensors is the base of dead reckoning. The analysis of step detection and step-length error of each walking pattern is shown in Tables 1 and 2. Comparing the detected steps with the true steps, the error rate was no more than 8%. The error of multi-pattern-based step-length was no more than 4.84% compared with the true distance. Thus, the proposed multi-pattern-based step detection algorithm and the step-length model can obtain accurate estimation results.

**Table 1.** Error of multi-pattern-based step detection.

| Walking Pattern | True Steps | Detected Steps | Misclassification Steps | Error Rate |
|---|---|---|---|---|
| Forward | 100 | 98 | 2 (Not detected) | 2% |
| Backward | 100 | 95 | 4 (Forward), 1(Not detected) | 5% |
| Left Lateral | 100 | 92 | 5 (Forward), 3(Not detected) | 8% |
| Right Lateral | 100 | 93 | 4 (Forward), 3(Not detected) | 7% |

**Table 2.** Error of multi-pattern-based step-length.

| Walking Pattern | True Distance/m | Detected Distance/m | Error Rate |
|---|---|---|---|
| Forward | 50 | 48.62 | 2.76% |
| Backward | 50 | 48.34 | 3.32% |
| Left Lateral | 50 | 47.58 | 4.84% |
| Right Lateral | 50 | 47.91 | 4.18% |

The multi-pattern-based location updating algorithm can obtain real-time location information when the pedestrian executes different walking patterns indoors. A traditional dead reckoning algorithm can only be used in case of walking forward. In the case of walking backward, left lateral walking, and right lateral walking, the multi-pattern-based location updating algorithm is used to minimize the coordinate updating error due to the misjudgment of step detection. By using the multi-pattern step detection algorithm, different walking patterns are detected and the corresponding step length is calculated; then, location updating based on multiple walking patterns can effectively reduce the cumulative error of dead reckoning.

The walking path is shown in Figure 8a. The pedestrian started from the position of AP1 (0,0), walked forward to AP2 (0,10), right laterally walked to AP3 (10,10), walked backward to AP4 (10,0), and, finally, left laterally walked to AP1. The step-length model in Equation (6) was used, and the output frequency of location is 50 Hz was based on PDR only. In the traditional PDR method, the backward and lateral walking patterns can only be detected as a forward walking pattern; thus, the forward

walking pattern and mixed walking pattern are compared in this work. A set of forward walking samples were collected with the same walking path for comparison. The positioning result is shown in Figure 8b.

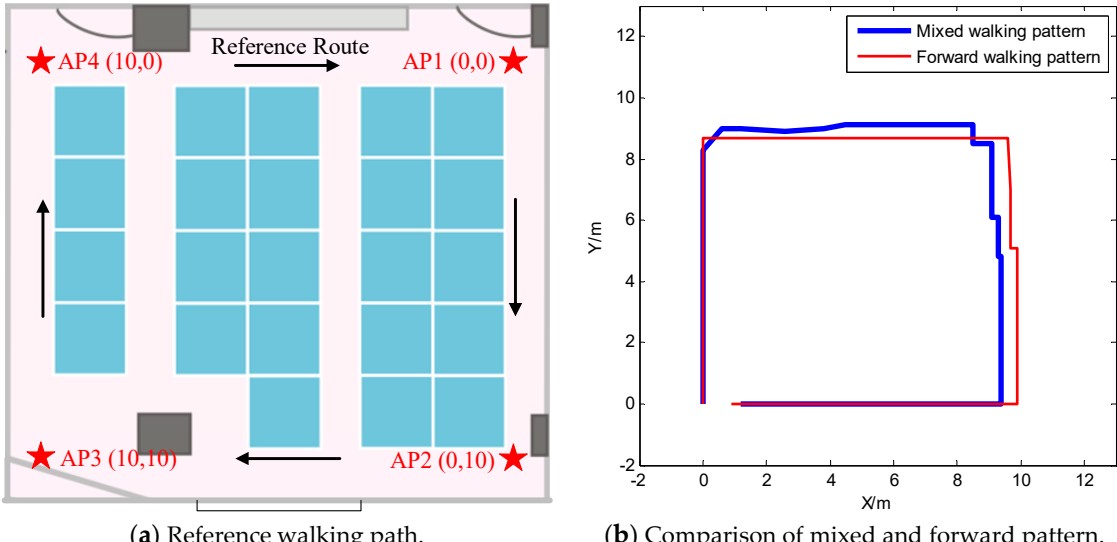

(**a**) Reference walking path.  (**b**) Comparison of mixed and forward pattern.

**Figure 8.** Comparison result of multi-pattern and forward-pattern.

In order to estimate the short-term accuracy of the multi-pattern-based PDR algorithm, 20 sets of data were collected using the same walking route and corresponding pattern for testing the closed loop accuracy of the multi-pattern-based PDR algorithm; a mean error of 1.95 m was gotten by calculating the Euclidean distance from the end point to the starting point, as shown in Figure 9. The experimental results show that the proposed multi-pattern-based PDR algorithm provided high accuracy among a short time, and the single forward pattern had higher accuracy than the mixed pattern.

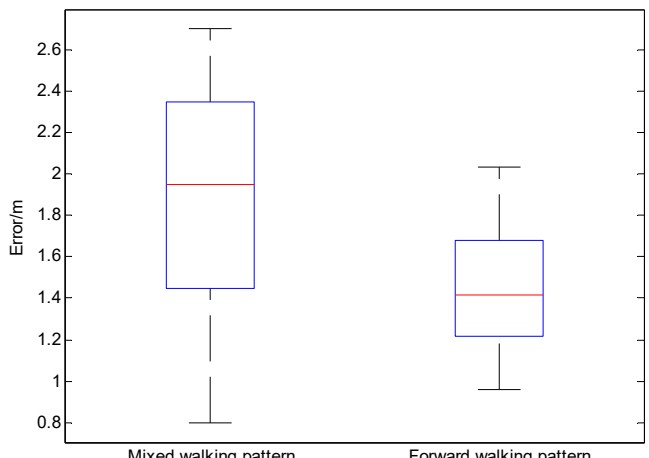

**Figure 9.** Positioning error of mixed-pattern and forward-pattern.

*5.2. Experiment Results of Wi-Fi FTM-Based Ranging Model*

The process of Wi-Fi FTM is affected by clock deviation, NLOS, and multipath propagation. The initial clock deviation can be estimated by the specific calibration method proposed in [14], and it can then be calibrated on an Android P-based Google Pixel 3 by subtracting the initial clock deviation from acquired RTT data. In case of NLOS and multipath propagation, a ranging model build in Section 3 can be used to limit the measurement error of Wi-Fi FTM. The experimental site containing a walking path is shown in Figure 8a, and the LS algorithm [27] was used to calculate the real-time

location. In the case of using the LS algorithm without a ranging model, the acquired RTT data will be affected by the ranging error mentioned above. In the case of using the LS algorithm with the ranging model, the ranging error is decreased, and better localization performance is realized. The comparison result is shown in Figure 10. The place marked by the triangle in Figure 10a is a pillar which could causes NLOS distance.

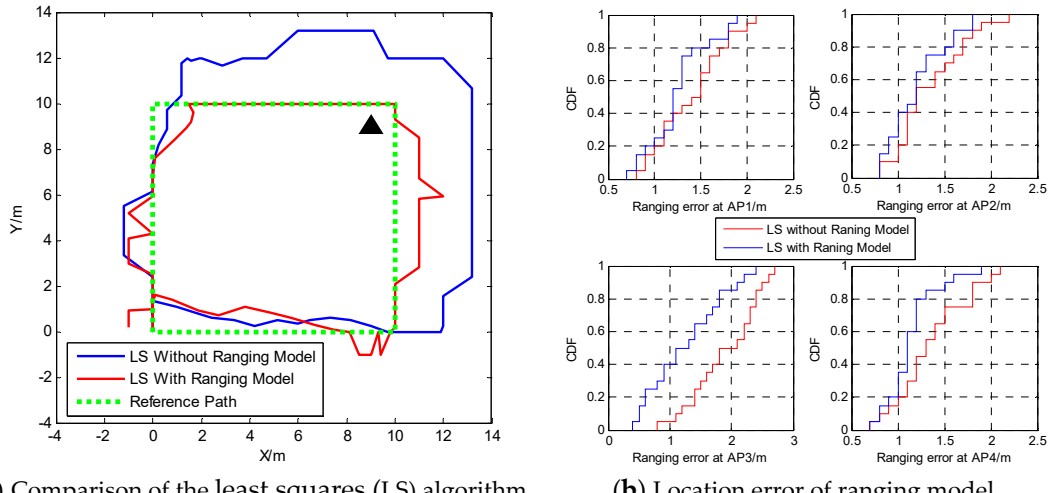

(**a**) Comparison of the least squares (LS) algorithm. (**b**) Location error of ranging model.

**Figure 10.** Estimation of Wi-Fi FTM-based ranging model.

In order to estimate the robustness of the proposed Wi-Fi ranging model in an LOS/NLOS mixed environment, 20 sets of data were collected using the same walking route in Figure 8a. Due to the occlusion of the pillar shown in Figure 7, the acquired RTT data at location of AP3 always contained effects of NLOS, while the locations of AP1, AP2, and AP4 did not. Figure 10b gives a comparison of the presence or absence of the Wi-Fi ranging model using the LS algorithm. The positioning error at each location of each AP was calculated by Euclidean distance. It can be found in Figure 10b that the proposed Wi-Fi ranging model effectively improves the accuracy of Wi-Fi ranging result, especially when there are obstacles blocking in the transmission path of the Wi-Fi signal. Thus, the NLOS-based error is effectively eliminated by the proposed ranging model, which contributes to the improvement of positioning accuracy.

*5.3. Experiment Results of DRWMs Algorithm*

Section 5.1 presented the accuracy of the multi-pattern-based PDR algorithm in a relatively short time. However, the PDR-based algorithm is not suitable for long-term localization due to the cumulative error of heading, step-length, and the location updating method. The Wi-Fi FTM-based positioning method has no cumulative error, but it is affected by NLOS and multipath propagation. In this paper, a multi-pattern-based PDR algorithm is proposed to eliminate error of step detection and location updating. Then a Wi-Fi FTM-based ranging model is established to minimize the ranging error caused by clock deviation, NLOS, and multipath propagation. Based on these, DRWMs are proposed to fuse the short-term accuracy of multi-pattern-based PDR and long-term accuracy of Wi-Fi ranging. In order to estimate the long-term localization performance of proposed DRWMs, a rectangular room was used as an experimental site, which is shown in Figure 7. The pedestrian started from the position of AP1 (0,0), walked forward to AP2 (0,10), right laterally walked to AP3 (10,10), walked backward to AP4 (10,0), and finally left laterally walked to AP1. The walking path is shown in Figure 8a. This process was continuously repeated 10 times to estimate the long-term accuracy of DRWMs compared with multi-pattern-based PDR algorithm. All experimental processes were continuous and there was no pause between walking procedure. The experimental result is shown in Figure 11.

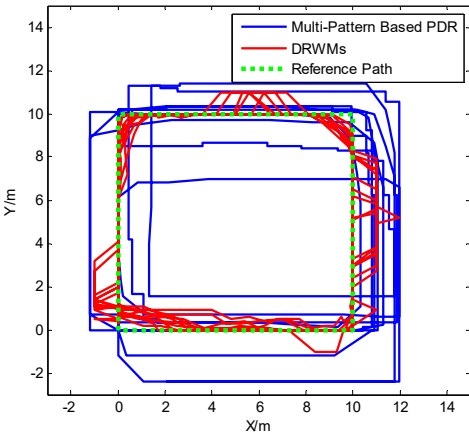

**Figure 11.** Comparison of multi-pattern-based pedestrian dead reckoning (PDR) and DRWMs.

In Figure 11, the proposed DRWMs algorithm, which is based on UKF, shows high accuracy and stability compared with multi-pattern-based PDR using the proposed step detection and step-length calculation method through long-term estimation. In the process of long-term measurements, the step-length of the PDR method causes the error of the final localization performance. It is foreseeable that with the duration of testing, the cumulative error caused by PDR method increases. Using the DRWMs algorithm obviously improves the long-term accuracy. We recorded the real-time locations when passing the reference point AP1 between the process of testing and got the comparison of position error by calculating the Euclidean distance between the real-time location and the position of the reference AP. Experimental results are shown in Figure 12.

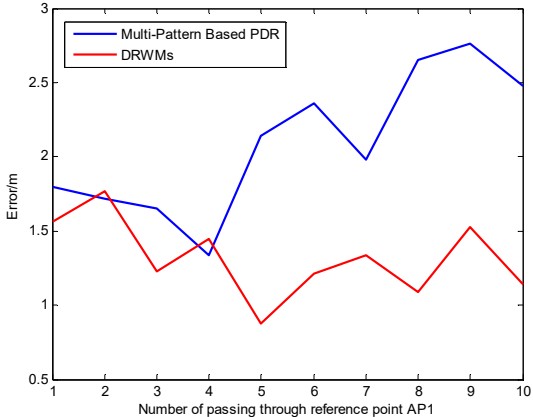

**Figure 12.** Comparison of error with long-term estimation.

It can be found in Figure 12 that after a long period of continuous testing, the error of PDR-based method begins increasing while the proposed DRWMs algorithm maintains high precision.

In order to estimate the expandability and stability of the proposed DRWMs, a rectangular office (12 m * 12 m) and a vertical corridor (45 m) was chosen as the first experimental site. The pedestrian started from the position of AP1 (0,0), walked forward to AP2 (0,10), right laterally walked to AP3 (10,10), walked backward to AP4 (10,0), left laterally walked to AP1, walked forward out of the office, walked forward to AP6 (−34,−2), and finally returned to AP1—as shown in Figure 13. When the corridor could be seen as the one-dimensional environment, just two APs (AP5 and AP6) were available. Five APs (AP1, AP2, AP3, AP4, AP6) were chosen as reference points. 10 sets of data were collected using the same walking route by different people, and the corresponding walking patterns and positioning errors at each reference point are shown in Figure 14:

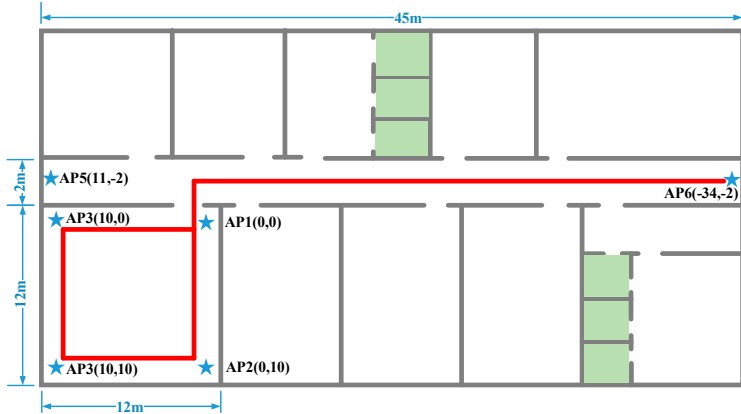

**Figure 13.** Experimental site contains office and corridor.

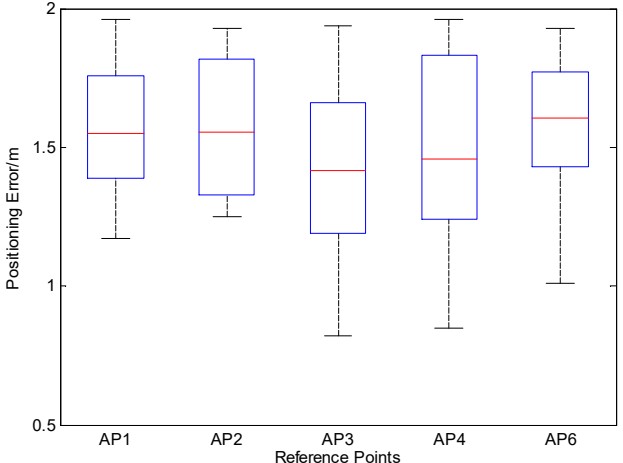

**Figure 14.** Positioning errors at each reference point.

Figure 14 shows that the maximum errors at each reference point are all within 2 m. This process was repeated 10 times to estimate the long-term accuracy and stability of the proposed DRWMs. All walking processes were continuous, and there was no pause between walking procedures. The total distance of walking was about 760 m, which took 10 minutes. The real-time location was recorded each time when passing the AP1, and the comparison of the position error was gotten by calculating the Euclidean distance between the real-time location and reference AP. Experimental results are shown in Figure 15.

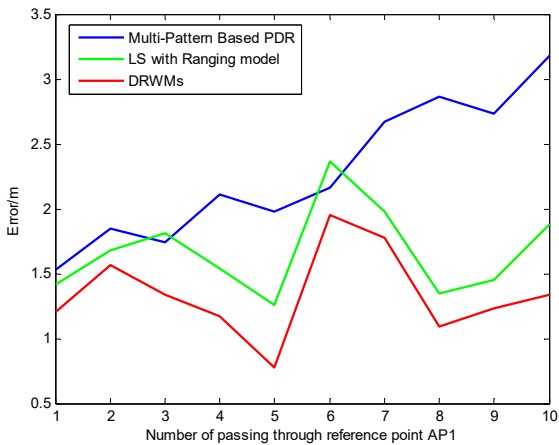

**Figure 15.** Long-term estimation in first experimental site.

Figure 15 suggests the high precision and stability of the proposed DRWMs algorithm compared with the single multi-pattern-based PDR and Wi-Fi FTM-based ranging model.

Then, a shopping mall was chosen as the second experimental site. The dotted line in Figure 16a indicates wooden shelf: The location of each AP shown in Figure 16a. The pedestrian started from the position of A(18,0), walked forward to B(0,15) and C(0,−5), left laterally walked to D(8,−5), walked backward to E(8,0), and then turned and right laterally walked to A(18,0). This process was continuously repeated 10 times, and the point E was selected as the reference point. The total distance of walking was about 780 m, which took 11 minutes. The real-time location was recorded each time when passing point E, and the comparison of position error was gotten by calculating the Euclidean distance between the real-time location and point E. Experimental results are shown in Figure 16b and indicate a higher precision and stability of the proposed DRWMs algorithm compared with the single multi-pattern-based PDR and Wi-Fi FTM-based ranging models.

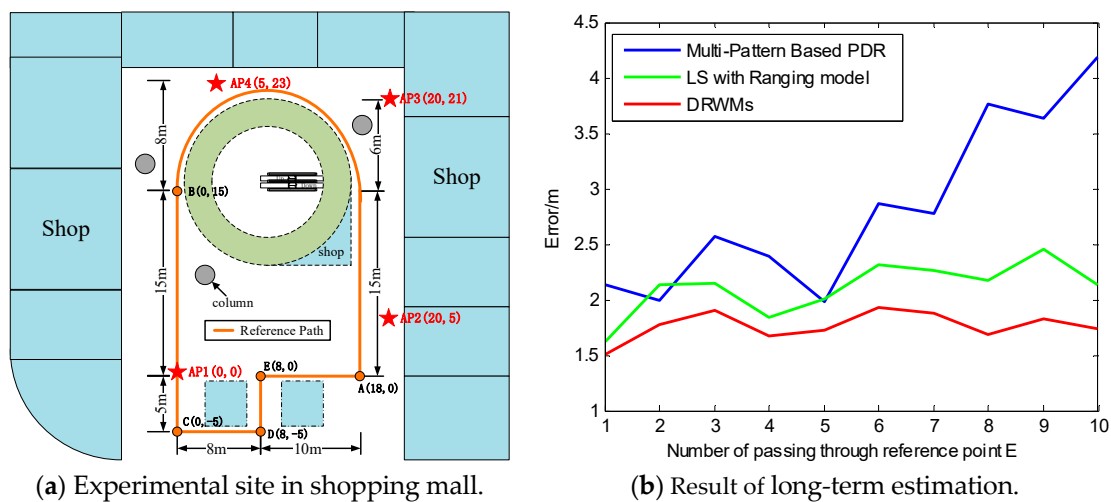

(**a**) Experimental site in shopping mall.　　　　　(**b**) Result of long-term estimation.

**Figure 16.** Long-term estimation in second experimental site.

Thus, the proposed DRWMs combine the advantages of the PDR method and Wi-Fi FTM-based ranging model and obtains more accurate location information. In general, the continuous improvements of the algorithm increase the localization performance.

## 6. Conclusions

To improve the accuracy and stability of indoor localization, this paper proposed the DRWMs algorithm, which is divided into three parts: (1) Multi-pattern-based dead reckoning via multiple sensors; (2) Wi-Fi FTM-based ranging model; (3) integrated localization using UKF. The initial location of UKF is provided by the LS method, and, then, dead reckoning begins. The step information contains walking patterns, and step-length is detected and calculated by the time-domain features of acceleration. The corresponding location updating algorithm takes different walking patterns into consideration and improves the flexibility and precision of multi-sensor estimation. The Wi-Fi FTM-based ranging model contains the effects of clock deviation, NLOS, and multipath propagation, which can effectively reduce the Wi-Fi ranging error and provides better localization performance. Finally, the multi-sensor estimation and Wi-Fi ranging model estimation are fused by UKF, and the advantages of two methods are combined to obtain higher positioning accuracy. The experimental results show that the proposed DRWMs algorithm achieves more precise and stable localization performance and satisfies different walking modes and indoor environment requirements. The final positioning error is within 2 m.

In the future, it is foreseen that the precise positioning results provided by Wi-Fi FTM and multiple sensors can gain further improvement by fusing multiple sources of information estimated from Wi-Fi

signal such as RSSI, AOA, and CSI. With more and more Wi-Fi chipsets in mobile devices supporting large bandwidth transmission—up to 160 MHz or more—the precision of RTT ranging Wi-Fi FTM will largely improve, which will be benefit to the high accuracy indoor localization.

**Author Contributions:** This paper is a collaborative work by all the authors. Y.Y. proposed the idea, implemented the system, performed the experiments, analyzed the data, and wrote the manuscript. R.C. and L.C. aided in proposing the idea, gave suggestions, and revised the rough draft. G.G., F.Y. and Z.L. assisted with certain experiments.

**Funding:** This research was funded by the National Key Research and Development Program of China (grant nos. 2016YFB0502200 and 2016YFB0502201) and the NSFC (grant no. 91638203).

**Acknowledgments:** Part of this work is supported by the Collaborative Innovation Center of Geospatial Technology, Wuhan University and provided the experimental sites and testers.

**Conflicts of Interest:** The authors declare that they have no conflict of interest to disclose.

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
