# Peer review of "A Robust Dead Reckoning Algorithm Based on Wi-Fi FTM and Multiple Sensors"

_remotesensing, doi:10.3390/rs11050504_

Round 1

Reviewer 1 Report

Wi-Fi FTM is a very hot topic at the moment, providing new (more accurate?) localization
approaches compared to well-established RSSI-based ones (models, fingerprinting).
The authors present a 2D indoor localization system for smartphones,
using PDR (with some enhancements) and absolute location information
provided by multi-lateration using multiple FTM distance measurements.
Both information are fused using recursive density estimation via UKF.

While the FTM topic is very relevant to actual real-world indoor localization
and navigation scenarios, the provided PDR extensions are rather theoretic.
Furthermore, the overall localization is just 2D and no floorplan is
considered during the density estimation. State of the art literature
uses a floorplan and 3D localization, often based on the particle-filter.
Also it is unclear whether the system supports tilt-compensation or requires
the pedestrian to hold the phone exactly parallel to the ground.

Abstract:
    "Wi-Fi Fine Time Measurement (FTM) protocol which can be used for Wi-Fi ranging"
    RSSI can also be used for ranging, but FTM should (in theory) be much more robust.
    Might be important for the reader

    "we propose a robust dead reckoning algorithm based on Wi-Fi FTM"
    This one is misleading. Dead Reckoning refers to relative updates,
    but FTM provides absolute location information.
    Point out its a combination of Dead Reckoning (Accel/Gyro) + FTM as
    absolute base to correct for (P)DR errors

    "can achieve accurate localization performance within 2 m."
    Such claims strongly depend on the used test-bed.
    It is almost always possible to design some test-bed to generate high
    accuracy/precision. Even with RSSI instead of FTM

    "Compared with the traditional Wi-Fi positioning"
    What do the authors refer to as "traditional Wi-Fi positioning"?

    "the proposed algorithm is more accurate"
    Is it? There are fingerprinting approaches with errors below 2m.
    At least, there are papers claiming such error margins.
    However, FTM should be much faster to set up which is a huge benefit!

L59:    "caused by clock deviation"
        what does this refer to?

L62:    "final dead reckoning"
        is it still dead reckoning if there are absolute sensors involved?

L70->71    sections (2. ..) usually start with an introduction, and not directly with a sub-section (2.1. ...)

L73        typo. "send" -> sends
L74        typo. "receive" -> receives
        or use plural for AP / Terminal

    In General:
    FTM sounds like simple time measurements. Wasn't this possible before FTM?
    Why does it require a new standard to perform them?
    -> Because this are very fine measurements, down to pico-seconds. Point out to the reader

L70        On does not write "equation (1)" but simply just "(1)"

L80        RTT must not be italics. Otherwise it means R times T times T
        Parentheses should use different sizes to make the order more clear
        The way the average RTT is described is not intuitive.
        1/n sum ( (t4-t1) - (t3-t2) )
        is much easier to comprehend

L85        "protocol excludes the processing time on the >>>Wi-Fi AP<<< by subtracting (t 3 (k) – t2(k))"
        (t3-t2) is the processing time of the client. not the AP?

L89        "per burst n is set as 30 "
        What does this imply on the measurement rate?
        How much time does it take to complete one measurement, when n=30 ?
        Measurement rate is crucial for indoor navigation
        What about multiple terminals?
        Does this still work when several terminals perform FTM with the same AP? (actual real-world scenarios!)

L91        equation totally messed up
        again, "RTT" must not be italic. Use different parentheses sizes
        why not refer to (1) ??
        nobody actually writes "*" for "times" within equations

L92        wording

L95     sentence starts with "While" but ends prematurely... there is no "but" or something similar
        is there something missing?

L98        "RTT" not italic
        this equation uses the same "n" as (1) and (2) did. this is totally confusing
        in (1) and (2) n was related to the number of repetitions.
        Here, n is related to the number of access points.
        Use a different variable

        Is the LS solution exact, or is there an error introduced due to the linearization? (subtract x_0/y_0 from every measurement)
        Is it "three or more" or "four or more" ? (due to the relative linearization)
        This is important!

        Is this equation actually correct?
        I think all D_RTT(..) are off by one and there is the refernce D_RTT(0) missing within every row of the matrix
        Just like the x_0 and y_0 that is present within every line

L105    typo. pedestrian->pedestrianS

L110    what is "vertical value" ? Which coordinate system is this in?

L121    What is the world-coordinate-system?
        What if the smartphone is held in a different way? tilted/rotated, etc ?

L129    why is there an opening case "{" but nothing mentioned?
        does this work when the phone is not parallel to the ground? (typical use-case!)

L132    m1 -> m_1 (subscript) and math-font
L133    the same

L137    why is there an opening case "{" but nothing mentioned?

L140    m_2 (subscript)
L141    the same

L143    use full page width. much better visibilty of the plot-data
        what is ax and ay? is this A_x  and A_y from above equations?
        Figure and equations/text do not match
        Captions not sufficient. What part of the figure shows walking "forward" and what part
        shows "backward" ?
        What are the big red dots?
        What are the red stars?
        What are the small red boxes?
        Filtered? Using what kind of filter-setup? Moving average? How large? What sample rate?
        All details are missing!

L147    what is s times t times e times p underscore times l ....???
        do NOT use italics in equations for text!
        Again: What is "vertical" acceleration? Which coordinate system? What axis is this?

L150    do not use italics in equations for text

        In general:
            How robust is this kind of step length detection?
            What are the constraints for this to work correctly?
            What if the smartphone is held non-parallel to the floor?

L152    Just like with "traditional Wi-Fi" what is "traditional Dead Reckoning"?
        The authors must explain what there understanding of traditional is!

        In general:
        question that comes to mind:
            How often does backward and lateral walking occur during typical indoor localization/navigation scenarios?
            Are these actual real-world use cases?
        Much more relevant real-world question:
            How does the multi-pattern detection react, what the pedestrian moves/shakes/tilts the smartphone while walking?
            Does this work out?

L171    I can't check this equation. Its totally distorted
        Again: Why is there an opening case "{" ???

L172    What is "initial heading" where does this one come from?
        What if the initial heading is unknown? (again: typical real-world scenarios!)

L173    "S_k is the flag of detected walking patterns"
        What does this mean? what kind of flag?

        In general
            So S_k is forward/backward/something-else
            and U_k = left/right/something-else
            What if the algorithm sets S_k != 0 and U_k != 0 at the same time?
            Like: "forward + left" can this happen? Must this be suppressed?

L182    "by detecting step AND HEADING"
        how is the heading detected? This is nowhere described
        Does this mean, there is no heading detection?

L185    Does Wi-Fi FTM actually use real time measurements based on clocks?
        Maybe it is indirectly inferred using phase information and there is no actual time-measurement?
        This is nowhere mentioned within the paper!
        If FTM uses clock information, is there actually a requirement for clock-synchronization?
        The AP measures time from send to receive using its clock, and also receives the time the Terminal needed for processing
        (calculated by the terminal itself)
        So there seems to be no need for time-sync at all?!
        Why is this problem mentioned? totally unclear
        Do [27] and [28] actually refer to FTM? or some totally different technique?
        Do [29]/[30]/[31] refer to FTM or some totally different technique?
        in FTM, the time and distance estimation is returned by the HW.
        Is there a kalman-filter inside of WiFi devices?
        [38] and [39] are from 2014 and 2012. I doubt they cover FTM topics? But they are cited as FTM within the text!

L203+    Both statements are rather broad... This totally depends on test-bed, setup, use-case, etc...
        Such statements can not be generalized!

L205    Again: Missing introduction for 3. .. 3.1. directly follows

L215    RTT non italic (just like the "true")
        what is t_4_n ? was this introduced? same for all other variables
        Align lines by the "=" sign

L218    Again: what kind of clock error? What clock?

L221    \Delta trandom << correct

L223    no italics for text
        what is D_true ? was this introduced
        What is C ? light speed? introduce all variables!
        The presented notation does not look like "zero mean". explain

        more intuitive notation would be something like
            X_{\Delta t} \sim N(0, \sigma^2)  where \sigma = ....
        X is a Gaussian random variable with zero mean

L228    typo: measurementS

        What can be corrected by this calibration? Is it t_delay ? is it t_random? both?
        This will only work to some degree! otherwise there would be no need for additional kalman filtering
        explain

L232    What does "mobile terminal" refer to? Laptop? Smartphone? Tablet? Embedded Hardware?
        Important!

L235    typo: deMoted

L237    why is "n" now referred to as an error distance? Previously "n" was used as a numbering variable for number of APs and number of Measurements...
        do not re-use variables for various purposes! this confuses the reader
        What is L_i ?
        what is i?
        again: no italics for text
        What is || .. || ? Euclidean distance? introduce within text!
        P is a location.. but what kind of? (x,y)? (x,y,z)? <<< IMPORTANT AND UNCLEAR!
        And: Uppercase letters are usually matrices, not vectors!
        Also: in (3) locations (x,y) where given by \vec{x}
        Use the same variable names if they state the same fact!
        Why does d_random refer to (10) but in (10) it was called \Delta t_random?
        If its the same thing, use the same variable!
        confusing!

L240    what is "drandom" ?
        What is "b_i" ? where does this com from? what does it describe? where is it used?

L241    why is d_random omitted? what are the consequences?

L247    "LS [37] algorithm can be used to solve the above problem"
        What actually IS "above problem"?
        It is nowhere mentioned what the authors are trying to solve in the first place

L249    where does "i" start?

L250:    again, what is b_i?

L255    What is the difference between ||..|| and ||..||^2 ?? This is nowhere introduced
        typo: ki -> k_i

L259    non-bold A is a scalar value, not a matrix. use correct notation
        Again: is P a vector? then use p not P!

L262    Formulation. Sentence starting with "while" but ending prematurely.

L281    Why UKF and not EKF or Particle-Filter? Reason?

L284    Figure 6 suggests that there is not tilt-compensation for the smartphone? Is this correct?
        if so, the text has to mention this limitation

        What is "initial location"? Location at time t=0?
        if so, where does "initial location" come from?     What if it is unknown? (real-world scenario)
            EDIT: This is mentioned later.. but this is much too late!

        Caption: "Frame"? What frame? Framework?

L290    Equation strangely compressed
        What is X(t) ?  does it return a matrix or a vector? If vector, user lowercase letters!
        but: x was already used as position variable earlier. so you can not use that again

L291    So the system performs 2D localization and not 3D? Just guessing. nowhere mentioned.
        Why is "L" now used as a function? Previously it was a variable covering something completely different
        do not reuse letters/variables for different aspects! No one can follow this!

L292/293    Why do A and B use different fonts?
L293    is B a matrix or a vector? For vectors use lowercase letters in bold

L294    Sk Uk -> S_k U_k

L300    Z(t) returns a vector -> use lowercase bold z!
        is u a scalar value? it is not bold! confusing!
        on can not add a scalar value to a vector.
        what is x_i and y_i? Position of AP?
        It was earlier, but the reader can't know for sure, as the authors re-use variable names all the time

L301    If H is the unit matrix, why is it needed?

L304    Even though Kalman filters are well known, there are various different notations used for state, prediction and update.
        The authors must introduce the notation they are going to use for the individual steps!

L307    What is the intention of representation?
        Vector? Matrix? What is X^(i) ? What is "i" ?
        What is \hat{X} ?
        Why is "n" now re-used as dimension of the problem? Previously it was some kind of error metric
        What is lambda exactly? Barely mentioned

L308    Why is P now a covariance matrix? Previously it was a 2D position vector?
        Nobody can follow this while reading... >>>>Must<<<< be fixed!

        Once and for all: Make clear
            - what is a vector
            - what is a matrix
            - do NOT re-use variables for different use cases
            - DO re-use variables if they mean the same thing

L313    "i" now clearly is a loop variable...
        previously it was something else? Or wasn't it? make that clear!

L320    previously x(t) was "x at time t"
        now there is the notation x(t+1|t)
        What does that mean? It was not clearly introduced within the text

L323    what is "n"? The dimensionality? or something new again?
        what is w^(i) ?

L324    what is the z_t z_t in subscript of P?
L325    what is the x_t z_t in subscript of P? position x at time t? something different?
        re-using variables for different purposes

        !!!!!!!!!!!
        In General:
            While the presented WiFi FTM localization provides a single scalar value as position,
            the corresponding uncertainty is clearly non Gaussian!
            All sorts of Kalman filters are therefore not well suited for this kind of problem.
            Particle-Filters and similar are the way to go here.
            Even if is approximated to be a Gaussian, its size and shape greatly varies
            depending on errors and AP placements.
            The authors must provide a textual description, how their setup deals with those facts.
            The reader must get an appropriate impression on the way the problem is solved
            and what the trade-off of using the UKF is.
        !!!!!!!!!

L332+    "Multi-sensor based dead reckoning have better timing recursion which could be used as the state equation."
        What does this mean?

L345+    The presented test-bed is small. One AP exactly within every corner within a small room
        ensuring visibility all the time is fine for a synthetic setup, but not for actual
        real-world use cases.
        The "2 meter error" mentioned within the abstract should thus be removed as it is not representable
        and misleading.

L360    Ok, so the number of steps is well-detected
        But, how often is there a pattern misclassification? (Detected: forward, but was: backward, etc?)
        What about the introduced step-length detection? How well does it work? evaluation?

L361    Layout, table is split among pages

L362    If it is "most important", add corresponding evaluations to prove its effectiveness.

L365+    This is not an evaluation paragraph. Its the same as already mentioned earlier.
        Evaluate the number of misclassifications.

L371    What kind of coordinate system is this, if AP1 (upper-right) is (0,0) ?

L371+    If i understand correctly, the system is evaluated against itself?
        "How well does it work when just using forward"
        "how well does it work when using other walking patters"?
        Comparison against "traditional" systems just supporting forward walks?
        Is there forward detection better?

L376    Why are there mainly 90° angles during the walk? This seems unbelievable!
        The lines are much too straight for actual heading estimations.
        And no pedestrian is able to walk this straight.
        Were there any hacks?
        Was the step-length evaluation active or not? This is nowhere mentioned.
        Was the Kalman Filter active or not?
        How often is a location estimation performed? After every step? Crucial information missing
        Is there an ground-truth data to compare against?
        Overlay the paths on top of figure 7 to include architecture ??

L380    what is the "European distance" is this a new metric?

L382    What about more realistic walks for several minutes throughout a complex building?

L386    effected -> affected

L386+    Many aspectes were already mentioned earlier in exactly the same way. this is not an evaluation.

L387    Were the mentioned calibrations actually added to the presented system?

L394    Why are we sure that the "error is decreased" ? There is no ground truth to compare against
        There are no actual measurements. The reader just sees 2 paths, not knowing what the
        actual walk should look like. This is not scientific.

L399    "It can be found in Figure 10 that without Wi-Fi ranging model,..."
        NO! It can't be found. Crucial ground-truth information is missing. Either of the two presented results
        might be correct, or non of them. The reader does not know for sure.
        In Fact: regarding figure 7 and the mentioned scale of 15x15 meters, the blue path actually
        looks more correct and the red one too small as it would walk through obstacles shown
        in figure 7!
        The reader can not trust the results the way they are presented.

L404    typo presents -> presented

L405    "Experimental results prove the great localization performance compared with the traditional PDR method."
        No. They do not.
        There wasn't a comparison against "traditional PDR"
        So, where does this statement originate from? Prove?

L419    Again, the angles seem too rectangular.. Was there some thresholding?
        How was the walk performed? Stopping at each AP, turning, starting to walk again?
        Was the step-length evaluation active or not? This is nowhere mentioned.
        Is the blue one without the kalman filter and the red one with Kalman filter?
        This is not clearly describe.
        Always describe how the data was gathered and processed before it resulted in what can be seen.
        Again: the red path seems to narrow to fit Figure 7 and 15x15
        Overlay the paths on top of figure 7 ?? Add correct scales to Figure 7?

L424     "just drift about 20 degrees"
        Neither of the both results shows a clear heading drift.
        They both seem to be axis aligned.
        While the rectangular pattern should be kept, it should be rotated by several degrees.
        This is not visible at all. The results seem unbelievable and the Text "drift about 20 degrees"
        proves that there is something wrong with the figures.
        Where are those 20°? I can not see any rotations over time within the figure. They must be somewhere

L427    typo Euclid -> Euclidean
L427+    Add the locations of the 4 APs within Figure 11 and 10 or overlay both on top of figure 7
        This helps the reader to visualize the walk within actual surroundings.
        How was the error determined? The pedestrian will not have walked directly over the AP.
        So the ground-truth is erroneous as well. Was this compensated for?

L434    Add acutal scale (in meter) to the figure

L435    adding a straight corridor does not prove "universality"
        The placement of the APs is still very synthetic.
        Under real-world conditions they will be placed completely different and there will be less transmitters
        than used within the test-beds.
        The walk is synthetic as well. There are only 90° turns

L444    Just the overall system is evaluated. As it uses novel components, Each component
        should be evaluated on its own for this longer walk:

        error for:
            - just step/turn detection
            - just Wi-Fi FTM
            - both combined

        Otherwise the reader can not comprehend whether the presented components make
        sense and actually help to improve the localization process

        The walk is too short for the dead-reckoning to produce high error rates.
        The FTM starts to shine only after much much longer walks.
        So the setup does not match the authors intention.

L446    How long was this walk? 60 seconds?
        Were all 10 walks conducted by the same person?

In general:

    For all walks:
        How was the initial position determined?
            EDIT: this is mentioned in the conclusion?! why there? much too late ?!
            EDIT2: How erroneous was this initial location estimation? Ground truth?
        Somewhere in the text it is mentioned, that it is initialized via FTM
        This can't be the case for the setups were just step/turn is used.
        Was the initial position given exactly? (Error of 0 meter?)
        This is also crucial for understanding the presented results
        and for getting an impression of validity for more realistic real-world scenarios

Some sources are missing information, such as the year for [7][13](not explicit) [14],[30]

Author Response

Dear Reviewer: 

Thank you for your comments concerning our manuscript entitled “A Robust Dead Reckoning Algorithm Based on Wi-Fi FTM and Multiple Sensors” (ID: 447028). Those comments are all valuable and very helpful for revising and improving our paper, as well as the important guiding significance to our researches. We have studied comments carefully and have made correction which we hope meet with approval. Revised portion are highlighted in the paper using “Track Changes” function. 

       Special thanks to you for your good comments!

                                                                                                      Manuscript authors

Reviewer 2 Report

A brief summary

The paper deals with a topic of high interest in the area of Rf-based indoor positioning system. The paper's main contribution is the proposed algorithm with the goal to improve the localization accuracy of human walking in an indoor environment. Specifically, authors claim that the positioning error due to the limitation of sensor-based dead reckoning and Wi-Fi FTM ranging due to NLOS could be mitigated by fusing both measurement data together with unscented Kalman filter (UKF).

Broad comments

Although the contributions are clearly stated in the introduction, the detail explanation of methodology especially in the multiple sensors related sections are quite difficult to read and must be improved. The format and structure of equations such as subscript and notation are not well-organized and difficult to read such as in equation 8. Although plenty of literature is fairly cited in the introduction, they are barely utilized in section 2 and 3 in order to support the novelty of the author's proposed algorithm. The experiment procedures are fairly described but lack the detail on the experiment configuration, especially data collection.

Specific comments

The following are the lists of reviewer's comments that need to be carefully addressed.

1. Inappropriate use of the technical term. the authors are advised to be more careful when using the technical term and acronym.
   1.1) line 34: "I" in RSSI stands for "indication" (not "index"), line 35: "S" in CSI stands for "state" (not side). please refer to [10] in the introduction
   1.2) line 34-35 TOA, AOA, CIR could not be explicitly extracted from Wi-Fi signal but they can be technically estimated/derived from wireless channel or CSI. please refer to [8-9]
   1.3) line 44: the error in NLOS is not due to ranging mechanism but the lack of LOS path itself. Please refer to [15] 2nd paragraph of the introduction
   1.4) line 92: "speed of light" is more commonly used than "the speed of radio wave"
   1.5) line 186 and 213: TOA and DOA are not technology but it should be categorized as method/technique or algorithm in wireless communication. Please find [27-28] for references.

2. Insufficient and unclear explanation.

    2.1) Although authors stated at line 50-51 that there are various methods for data fusion, no reason or justification of why authors have decided to use UKF on the proposed DRWMs algorithm has been addressed.

    2.2) On page 4, it is difficult to understand how the proposed multi-pattern model works based on the descriptions at line 129-142. Please try to paraphrase or simplify paragraphs with simple sentences and avoids using multiple commas within a sentence. In addition, authors are encouraged to explain the physical behavior to the acceleration patterns if possible. At a very least, this section should include the clarification of how authors have come up with this method or it was due to pure observation.

    2.3) the advantage of the multi-pattern based location updating model in equation 6 should be comparatively justified with literature. For example, at what aspect the use of the proposed model is better than others.

    2.4) According to figure 6, there is no explanation in the previous sections of how gyroscope and magnetometer are applied in the DRWMs Algorithm. Reviewer assumes that the sensors should be used for the estimation of walking direction, specifically to calculate theta_k in equation 8. With this regard, authors are advised to elaborately explain the contribution of gyroscope and magnetometer to the algorithm. If authors apply techniques from literature, it needs to be clearly mentioned and cited.

    2.5) According to equation 12, could authors elaborate on the definition or physical interpretation of the NLOS error n_i? and how it is different from d_random? These detailed are recommended to add to the manuscript since it plays an important role in the model's derivation.

    2.6) As it is mentioned in line 387, authors used the specific calibration method from [14] to calibrate initial clock deviation \delta t_delay. Does this clock deviation mean the offset in [14]? If yes, what method the author used for calibration as [14] has specified two methods: 1) subtracting that measured offset from the readings or 2) by canceling this offset through the delay of long enough cables. If 1) method was used, for clarification, the figure showing the hardware setup at Wi-Fi APs should be added.

    2.7) Authors used Intel 8260 Wi-Fi chip which is the same version as the WiFi FTM Linux Tool developed by [14] http://www.winlab.rutgers.edu/~gruteser/projects/ftm/index.htm. If authors also used this tool, it is highly recommended to mention and cite in section 5. If the software was custom-made by authors, it also should be clearly stated.

3. Experiment-related comments

    3.1) The content structure of the experiment results could be slightly modified. Typically, the overall experimental scenarios and setups should be summarized at the beginning of the section. Hence, authors are advised to describe the experimental scenario in figure 13 before section 5.1.

    3.2) Authors have not provided the detail of data collections. For instance,
         - how many IMU sensor and Wi-Fi FTM data have been collected per second (sampling frequency)?
         - Are both sensors collected with the same sampling frequency? and how authors synchronize their timestamp if both data are collected separately.
         - where the proposed positioning/tracking is processed (smartphone, AP, or PC), and is the algorithm run real-time or offline?
         - what software has been used to collect the IMU sensor, Wi-Fi FTM ranging data, and process the proposed algorithm?

    3.3)  The result in figure 8 has shown the walking route to be 8 meters in y-direction which is contradicted with the explanation at line 371-372 at figure 7.

    3.4) The ground truth of walking course should be added to figure 9 for visual comparison.

    3.5) In section 5.2, the main advantage of the proposed ranging model is the robustness in NLOS scenario. Therefore, authors should evaluate the performance of the ranging model in both LOS and NLOS possibly in terms of CDF localization error. Authors may find [2,14] for reference.

    3.6) What is the reason that authors have selected the location near APs for the evaluation in Figure 12 and 14? Usually, in positioning-related research, the test points (or reference point in this context) should be selected randomly throughout the experiment scenario as showed in [14].

    3.7) Although line 435-439 provided the 2D-coordinate of 6 APs, there is no scale to refer the coordinates to figure 13. At least the total size of the office and corridor should be added in the figure.

Author Response

(The authors gave the same response as above.)

Round 2

Reviewer 1 Report

Major improvements were made by the authors.
However, some aspects are still unclear or incorrect (see comments)

If the setup is intended for competing against state of the art indoor localization and
navigation systems, the experiments should be refined. The provided test-setups
and walks are rather synthetic, and tailored to produce good results.

L76+     " In addition, RSSI based Wi-Fi positioning method has poor real-time performance, it may take one or more seconds to get a new location."
    Scanning for nearby APs can be as fast as 600ms when using the 2.4 GHz band only. There is literature for that.
    Furthermore, one 600ms scan can list dozens of nearby AccessPoints.
    For FTM, one hast to explicitly specify the MAC-addresses of the APs to measure, and the number of batch-scans is limited in Android 9.
    This is unfeasible for large buildings with dozens of APs. So the statement is misleading and needs correction / further explanation.

L78+ "we can get much higher update rate"
    how fast is "much higher" compared to  " it may take one or more seconds"
    quantify for comparison

L79+ "and fine measurement result which is down to pico-seconds."
    This is misleading and sounds like "one measurement takes just pico-seconds"
    Furthermore, related to the previous sentence, this sounds like a comparison
    between RSSI and time-measurements.

Previous comment on L89
    "per burst n is set as 30 " What does this imply on the measurement rate?
        The authors sent an explanation in the comments. But was it also added to the paper?
        This kind of information is important to the reader, as it concerns real-world use and limits
    Comment on the comment:"such as the speed of the processor and bandwidth"
        Is the processor really part of the equation? Is there any proof for that?
        Measurements down to pico-seconds can not be performed by normal CPUs as their
        clock rate is far too slow. There must be some sort of dedicated circuit for this one.
        Can you add proof/details or refer to other literature for proof?

L105    in the received PDF, the equation is much better, but still visually corrupted
        brackets are displayed within the variables etc.
        Maybe this is a PDF issue?

Comment on comment 19 "Is the LS solution exact..."
    "it can get optimal estimated location information by measured distance between
    terminal and each AP, when there are three or more APs, 2D location information can be calculated"
        yes, it "can", but usually does not.
        See: https://ieeexplore.ieee.org/document/1440903
    "when there are three or more APs, 2D location information can be calculated"
        does the LS-linearization affect the number of required transmitters? explain

L140 "Texting pattern"
    Texting pattern is usually considered using a tilt of 30-50°
    Does the system still work for these angles?

L153    "The sample rate is 50Hz"
    add information which sensor this refers to: Accelerometer?

L201    equation is visually corrupted in the received PDF

L202    "which is provided by magnetometer"
        The magnetometer is often very erroneous and sometimes completely unreliable.
        Furthermore, it requires for correction depending on the location on earth
        to correlate magnetic and geographic north.
        How is this addressed?

Comment on 38:
    There still needs to be a proof or cite that clearly states how FTM determines the time of flight.
    The same goes for the mentioned "clock errors":
    According to (1), there can be no clock error that needs synchronization or calibration,
    as the equation removes any deltas.
    " Wi-Fi devices do not contain a kalman-filter"
    I know that they do not. But the text reads as if they do.
    If there is something as "Random clock error", it occurs directly within the PHY hardware,
    requiring the PHY to contain a Kalman filter, which it does not.
    The given sources relate to TOA and DOA, yes. But they describe different techniques to determine
    the TOF which DO require filtering in software. They are not related to what FTM does.
    This must be made clear for the reader

L246+ "But the raw data from FTM contains clock deviation error"
    Again: Missing proof/cite for this one. According to (1) there is no "clock deviation".

Comment on 47:
    laptops support FTM, but to not contain an IMU.
    My question had the following in mind:
        For any scientific research, setups and experiments must be explained in a way,
        that others can reproduce them. Either for validation or falsification.
        This includes used hardware, sensors, sample rates, filters, quantification of all parameters and variables,
        detailed explanation on the test-bed, and so on.
        The reader wants to know the details on everything that is required for replicability.

L286
    " P = [x 0 ,y0]T indicates the location of mobile terminal, Pi is the location of Wi-Fi AP, || P - Pi || is the Matrix norm
    if P=(x,y)^T, then P is a vector, not a matrix.
    Pi is the location of Wi-Fi AP, again, P seems to be a vector.
    The difference P-P_i of two vectors is a vector as well.
    So: why is ||..|| a Matrix norm?

Comment on 51
    "Because drandom is always between -0.5m to 0.5m according to the [14]"
    This must also be explained to the reader, not only to the reviewer.
    Otherwise the reviewer will understand, but non of the readers does.

Comment on 63: "B is a matrix, indicates .."
    make clear for the reader

Comment on 75:
    It is not only the non-gaussian random errors.
    Much more important:
    The whole probability density for lateration is non-Gaussian, most of the time,
    depending on how the circles around each AP are intersecting, the shape completely changes.
    This non-Gaussian shape can not be modeled by any sort of Kalman filter.
    e.g. shown here: https://ieeexplore.ieee.org/document/7117991

Comment on 76:
    make clear to the reader, not only the reviewer

Comment on 77:
    " it can be much smaller in LOS contained environment"
    And it will most likely be much larger for real-world setups with many obstacles
    along the LOS and a reduced number of APs, when using existing Wi-Fi infrastructure.

Comment on 84
    \Delta h does not seem to be quantified within the work, so the reader can not
    understand its impact, and the amount of thresholding.
    If there is a threshold, what about real-world scenarios, where smaller turns
    are quite likely? What if a pedestrian slowly walks along a very large curve,
    a rounded corner, or within a rounded building? This will yield rather small differential heading changes.
    Will the result be a straight line, or will it work?

L472:
    sentence starts with "while"
    This reads as if the calibration from [14] was NOT used within this work?

Comment on 90:
    how is the CDF estimated? This usually involves many comparisons between
    estimation and real-world location, and thus a ground truth,
    which is not described within the work.
    Is solely the error at AP3 considered and used several times?
    If so, the result is very biased and barely useful for actual reasoning.

    The authors should definitely consider introducing ground-truth for much more reliable error estimations

Figure 10
    add locations of the APs
    add at least an expected ground truth, so the reader can verify the differences.

Figure 11
    the same as Figure 10

Comment on 97:
    Also make clear to the reader, not only the reviewer.
    Again: setups and results must be explained in detail.

Comment on 99:
    "figure.13 shows that the proposed DRWMs algorithm can be used in more scenes."
    This setup is still very synthetic and does not resemble real-world scenarios:
    - too many access points
    - too many straight walks
    - only 90° turns
    - just one room / corridor
    - APs are placed to yield good results. AP-placement does not resemble existing infrastructure setups

    The paper needs a more realistic testbed and walk for the reader to
    get a real impression on the quality of FTM and the proposed algorithms

L543:    "Also this process is continuously repeated 10"
    Just to be sure what "repeated" means:
    The process was NOT interrupted between the 10 repetitions, right?
    -> The walk was repeated, but the location estimation was running the whole time?
    Add: how long was this walk (in meter and seconds)
    This helps the reader

Author Response

Dear Reviewer: 

Thank you for concerning our manuscript entitled “A Robust Dead Reckoning Algorithm Based on Wi-Fi FTM and Multiple Sensors” (ID: 447028). Those comments are all valuable and very helpful for revising and improving our paper, as well as the important guiding significance to our researches. We have studied comments carefully and have made correction which we hope meet with approval. Revised portion are highlighted in the paper using “Track Changes” function. 

Special thanks to you for your good comments!

                                                                                                      Manuscript authors

Reviewer 2 Report

A brief summary
Manuscript has been intensively revised in terms of English language and explanation. However, the reviewer strogly recommend the manuscript to be proofread especially by the specialist editor in the field of wireless communication.

Broad comments

Regarding the report from the authors, reviewer is not convinced in the item 2.5. What the reviewer understand is that e_i and d_random (which derived from t-random) are both random variables due to ranging measurement but e_i only exists only in NLoS scenario. More research and measurement are needed to prove the existance of this e_i. Authors are strongly advised to cite the reference regarding the e_i if available. On the other hand, if the role of e_i is the auxiliary variable, then the author is advised to clearly explain which is more appropriate than the error due to the reflection.   

Specific comments

The following are the lists of reviewer's comments that need to be addressed.

1. Line 73-74, RSSI based Wi-Fi positioning method is much more dependent on the environment because of the multipath propagation" does the author refer to the RSSI fingerprinting based positioning system? If yes, then the statement is true and the authors are advised to be specific in the term "fingerprint".

For the author information, RSSI is highly interference dependent, hardware-dependent, also sensitive to environment factor such as temperature. Authors may refer to [1-3]

2. Line 77, do the authors refer the traditional dead reacking to PDR?

3. Line 137 what do texting pattern and subtle mean in this context?  

4. Line 138, reviewer suggests to avoid citing equation before it is mentioned.

5. Figure 8, the performance of positioning tracking seems to be larger in the area between AP1-AP2 in both cases (visually observed from the small flucuation and the walking range is around 1 meter shorter than groundtruth). could the authors elaborate on this result?

6. Figure 12 and 15, label on x-axis should be revised. It is recommended to emphasize that the result is measured at AP1 to avoid the ambiguity. In addition, since both figures showed the same comparison for different environments, is there the reason why figure 12 does not include the result in Ranging (green line in figure 15)

references.

[1] Sandoval, R.M.; Garcia-Sanchez, A.J.; Garcia-Haro, J. Improving RSSI-based path-loss models accuracy for critical infrastructures: A smart grid substation case-study. IEEE Trans. Ind. Informatics 2018, 14, 2230–2240.

[2] Lui, G.; Gallagher, T.; Li, B.; Dempster, A.G.; Rizos, C. Differences in RSSI readings made by different Wi-Fi chipsets: A limitation of WLAN localization. 2011 Int. Conf. Localization GNSS, ICL-GNSS 2011, 2011, pp. 53–57.

[3] Boano, C.A.; Wennerström, H.; Zúñiga, M.A.; Brown, J.; Keppitiyagama, C.; Oppermann, F.J.; Roedig, U.; Nordén, L.Å.; Voigt, T.; Römer, K. Hot Packets: A Systematic Evaluation of the Effect of Temperature on Low Power Wireless Transceivers. Extrem. Conf. Commun. Association of Computing Machinery, 2013, pp. 7–12

Author Response

(The authors gave the same response as above.)
